# Generalized precursor prediction boosts identification rates and accuracy in mass spectrometry based proteomics

Aaron M. Scott [1✉], Christofer Karlsson [1], Tirthankar Mohanty [1], Erik Hartman [1], Suvi T. Vaara[2], Adam Linder[1], Johan Malmström [1] & Lars Malmström [1✉]

Data independent acquisition mass spectrometry (DIA-MS) has recently emerged as an important method for the identification of blood-based biomarkers. However, the large search space required to identify novel biomarkers from the plasma proteome can introduce a high rate of false positives that compromise the accuracy of false discovery rates (FDR) using existing validation methods. We developed a generalized precursor scoring (GPS) method trained on 2.75 million precursors that can confidently control FDR while increasing the number of identified proteins in DIA-MS independent of the search space. We demonstrate how GPS can generalize to new data, increase protein identification rates, and increase the overall quantitative accuracy. Finally, we apply GPS to the identification of blood-based biomarkers and identify a panel of proteins that are highly accurate in discriminating between subphenotypes of septic acute kidney injury from undepleted plasma to showcase the utility of GPS in discovery DIA-MS proteomics.

[1] Division of Infection Medicine, Department of Clinical Sciences, Lund University, Lund, Sweden. [2] Division of Anaesthesia and Intensive Care Medicine Department of Surgery, Intensive Care Units, Helsinki University Central Hospital, Box 340, 00029 HUS Helsinki, Finland. ✉email: aaron.scott@med.lu.se; lars.malmstrom@med.lu.se

Recent computational advances in the development of fragment spectra prediction[1–4], and the creation of full proteome repository scale spectral libraries[5–10] have allowed for the exploration of DIA as a means for discovery proteomics. Although these large libraries can help facilitate the identification of novel proteins in a DIA experiment, they present significant computational difficulties, particularly when attempting to control the false discovery rate (FDR) using the target-decoy approach[11]. The increased search space of these libraries can cause a decrease in sensitivity and statistical power as more false positives are introduced into the library, leading to less true positive precursors being correctly identified[12–14]. Attempts have been made to filter down these massive libraries to sample-type-specific libraries in a data-dependent fashion to create a more manageable search space for statistical validation algorithms[13,15]. However, if the library filtering is done too strictly potential true peptides are eliminated unnecessarily from the analysis, and if done too liberally many false positives can be left in the library for consideration. In both cases, this can result in an unreliable estimation of the FDR, so care must be taken when filtering large spectral libraries for analysis.

In addition to the filtering and sub-setting of spectral libraries, one area that has not been investigated with the same vigor is the effect of the choice of statistical validation algorithms on their ability to control the FDR in a stable manner. The method of choice for the validation of DIA mass-spectrometry data is the mProphet algorithm[16], which is similar to the Percolator algorithm commonly used in DDA proteomics[17]. The mProphet algorithm, implemented in the python package PyProphet[18] for the use of validating DIA data extracted with the OpenSWATH software[19], uses a semi-supervised method to combine all calculated subscores into a final classification score used to control the FDR. This method works by selecting positive training targets below a particular $q$-value cutoff (typically 0.01 or 1% FDR) in an iterative fashion until the number of identifications passing the defined threshold is maximized. In most cases, where the library closely matches the proteome of the sample of interest, this method works exceedingly well. Apart from the OpenSWATH tool chain, more modern approaches such as DIA-NN[20] utilize a modified version of the mProphet algorithm, where positive targets are selected based on an initial FDR cutoff and then an ensemble of neural networks are trained to classify target and decoy precursors. In addition, EncyclopeDIA[21] directly uses Percolator[17] to validate precursors, while MaxDIA[22] trains an XGBoost classifier for each experiment to control the FDR. All of these common DIA analysis tools utilize some aspects of the mProphet algorithm and rely on training new classifiers for each subsequent analysis, which can be a computationally heavy and nontrivial task.

In a typical blood-based discovery analysis, DIA-MS can be used to detect low abundant and tissue-leakage proteins that have not previously been detected using tissue-wide spectral libraries and has emerged as an important method for the identification and analysis of blood-based protein biomarkers. This type of analysis creates a large search space where the majority of precursors in the library are likely not contained in the sample, as blood plasma will not contain each of the tissue proteins included in the library. A large search space is also prevalent in proteogenomic experiments (i.e., searching for single amino acid variants) or the sequencing of antibodies from pull-down experiments, where the variation in potential protein sequences causes the search space to sky-rocket[23]. In these cases, "true" target labels are outnumbered by the "false" targets effecting the training of experiment and sample-specific classifiers and the accuracy of the resulting FDR control. Research has been done to develop methods that identify noisy labels and stabilize model training[24,25], but these are not implemented in common DIA analysis pipelines. In addition, if these "true" targets are identified they would represent a small fraction compared to the negative decoys in the data, creating an overwhelming class imbalance, which can destabilize the training of machine-learning algorithms if not dealt with in an appropriate manner[26].

In contrast to the established methods, we propose a generalizable machine-learning framework (GPS) that can be applied to any DIA-MS experiment to accurately predict precursors and control the FDR without the need to train new classifiers. This generalizable scoring approach has been demonstrated to work using static Percolator models in the context of DDA proteomics[17] but these models are trained on the sample types that they are used to evaluate, so it is unclear if they would generalize to diverse and unrelated data. We hypothesize that a good precursor is a good precursor, no matter the sample type, and that statistical validation models can be trained on unrelated external data if curated properly. These generalized models can be used to directly predict true precursors and eliminate false precursors from contention, allowing for stable FDR control regardless of the original search space. To that end, we have trained a generalizable scoring model and implemented a suite of algorithms to provide stable validation of extracted precursors through the search space size agnostic Generalizable Precursor Scoring (GPS) package (https://github.com/InfectionMedicineProteomics/gps).

## Results

**Overview of GPS.** The overall goal of GPS is to provide accurate FDR control regardless of the initial search space, while maximizing the number of proteins identified and providing high quantitative accuracy. The basis of GPS is to train models on curated data in an effort to maximize precision and then utilize these models to predict and score precursors in order to control the FDR (Fig. 1a). The high-precision classifiers can be directly applied to new data to predict true precursors from a sample and filter out low-confidence ones from analysis, allowing for stable FDR control independent of the original search space.

We evaluated GPS in four different scenarios described in detail in the sections below (Fig. 1b).

- First, we established a methodology for training GPS classifiers and demonstrate how they generalize effectively to new data, as described in "GPS effectively generalizes to new data".

- Second, we investigate how the predictive power of the generalizable classifiers can boost identification rates compared to existing methods while eliminating false identifications through entrapment FDR analysis in the section "Precursor prediction with GPS enhances identification rates".

- Third, we show how GPS can improve quantitative accuracy compared to existing methods in the section "Precursor validation with GPS improves quantitative accuracy".

- Fourth, we demonstrate the application of GPS through the analysis of novel blood plasma samples from sepsis patients with acute kidney injury (AKI) and use it to find potential protein biomarkers to effectively stratify two established AKI subphenotypes[27]. This analysis and results are described in the section "The application of GPS to identify potential protein biomarkers for sepsis-induced AKI".

This 4-part evaluation was performed using four distinct datasets (Fig. 1b).

- The first dataset, referred to as the yeast data, is a novel dataset consisting of 128 yeast samples run with varying

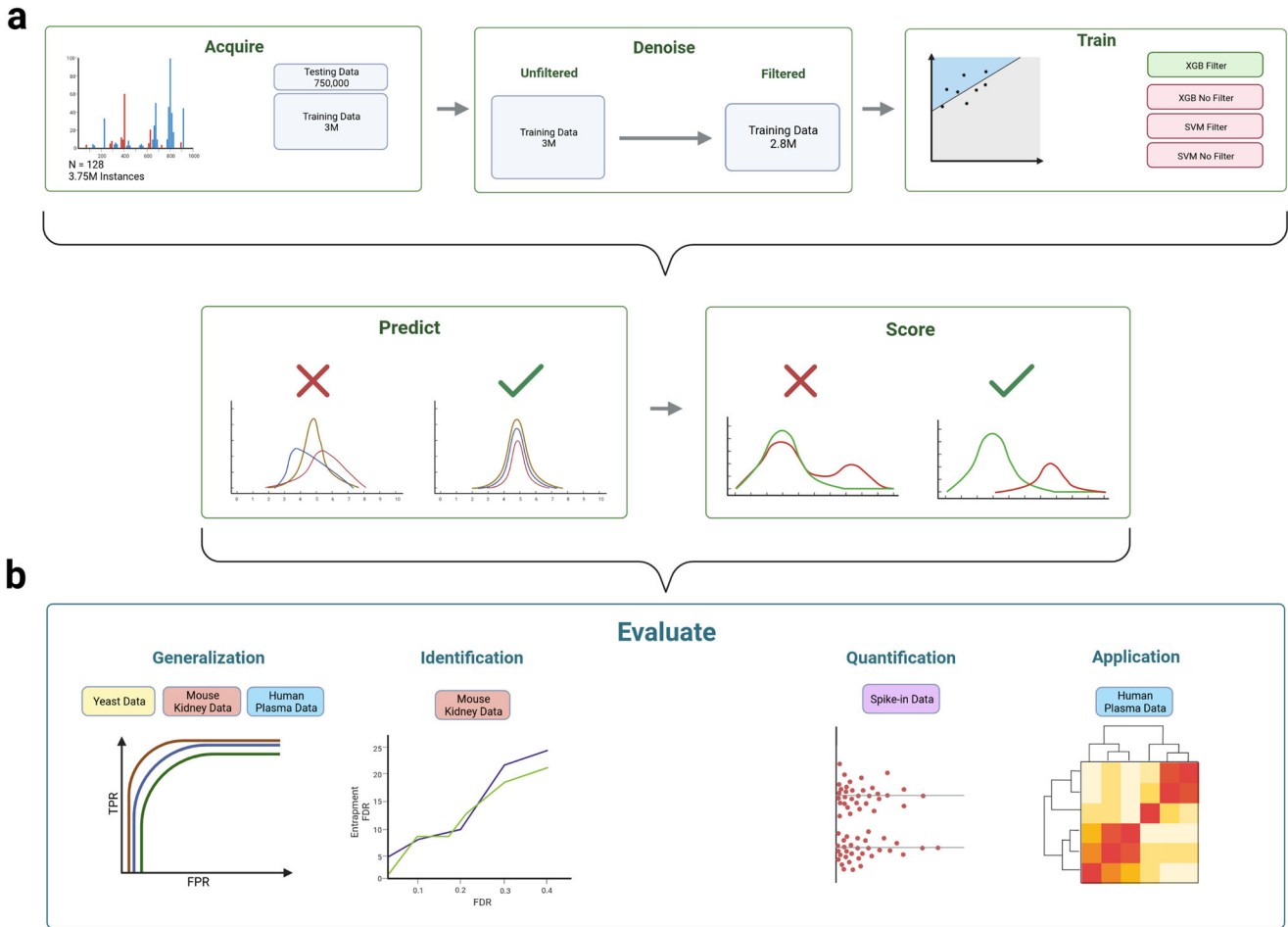

**Fig. 1 Overview figure depicting GPS and the methods and data used for evaluation.** GPS is first visualized in (**a**) and split into two groups. The first part of (**a**) visualizes the training procedure for the GPS models. The data from Yeast samples were first acquired at different gradient lengths from two different mass spectrometers and totaled 3.75 million precursors from 128 sample files. This data was then split into a train and test set. The training set was further filtered using a $k$-fold ($k = 10$) self-denoising algorithm where an ensemble of logistic regression models are trained for each fold and vote on the held-out data to determine the set of true precursors. This removal of false precursors results in a filtered training set of 2.8 million precursors. Two models, one SVM and one XGBoost, were then trained on the filtered and unfiltered training data for a total of four models. These trained models are then applied to new data to predict and score precursors to validate extracted signal in a DIA-MS experiment. **b** Visualizes the 4 separate methods used to validate GPS and compare it to existing methods, along with the data that are used for each analysis. To directly evaluate how GPS generalizes to new data, we measured the performance of the four classifiers on the yeast data, mouse-kidney data, and a subset of the human plasma data. We then measured and compared the identification rates of GPS and PyProphet using the mouse-kidney data in an entrapment FDR analysis. We then evaluated the quantitative accuracy of the validated identifications of GPS compared to PyProphet by analyzing a set of two-species mixture samples consisting of two groups and comparing the number of identifications that fall within the expected ratios. Created with BioRender.com.

gradient lengths (30, 45, 60, 90, 120 min). This yeast data provides a means to benchmark GPS using a simple and well-defined proteome that competing methods should also perform well on. The yeast data is used primarily to train the GPS models and a held-out portion of the data, the yeast test data, is used for evaluation in the section "GPS effectively generalizes to new data" below.

- The second dataset, the mouse-kidney (MK) data, consists of 31 mouse-kidney samples from a previous study[28]. These data represent a more complex proteome and is used to evaluate how the models generalize and how they maximize the number of identifications while maintaining a stable FDR in an entrapment FDR analysis. These data are used in two sections, "GPS effectively generalizes to new data." and "Precursor prediction with GPS enhances identification rates".

- The third dataset, the spike-in data, is a novel dataset consisting of tryptic Yeast peptides spiked into a constant

mouse-kidney proteome at known concentrations. These spike-in samples consist of two groups where 4X the yeast peptides are spiked-in to one group of samples so that we can measure the quantitative accuracy of GPS based on the expected ratios (0 log2 fold change for mouse precursors, and 2 log2 fold change for yeast precursors). This data and analysis is described in the section "Precursor scoring with GPS improves quantitative accuracy".

- The fourth dataset, the AKI data, is a novel dataset consisting of 141 previously unpublished blood plasma samples from a subcohort of sepsis patients with AKI from the FINNAKI study[27,29]. These 141 AKI samples are comprised of two established sepsis subphenotypes, based on the severity of the illness, that were developed from a combination of multiple clinical and molecular markers[30]. The AKI data are used to test GPS in a large search space where the proteins contained in the sample do not match up against the spectral library being used. From the AKI

data, we are able to identify more than 1300 proteins in undepleted human plasma and pinpoint a panel of the putative biomarkers to stratify subphenotypes of AKI using explainable machine learning. In addition, a subset of these samples are used to evaluate how GPS generalizes to human plasma samples. An overview of the four evaluation methods, and the sample sets used is visualized in Fig. 1b. A portion of this data is used in the section "GPS effectively generalizes to new data", and the data are analyzed completely in "The application of GPS to identify potential protein biomarkers for sepsis-induced AKI" below.

**GPS effectively generalizes to new data.** The first step in evaluating GPS is to establish a method for training models that will generalize effectively to new data. To that end, 129 samples of the yeast data were randomly split into 102 training samples and 26 test samples set aside to be used for validation and to assure the trained models are not overfitted. The combined data consists of 3,751,367 peak groups, while the 102 training samples consist of 2,988,116 peak groups (1,479,571 decoys and 1,508,545 targets), and we will refer to this data as the unfiltered training data. The test samples consist of 763,251 peak groups, and we will refer to them as the yeast test data. The unfiltered training data was filtered using a novel denoising algorithm to remove noisy false target labels that destabilize model training. This denoising algorithm takes a sample as input and initially splits the precursors in the sample into tenfolds. So that each held-out fold can be scored using classifiers trained on separate data, the remaining data is used to train an ensemble of ten logistic regression classifiers with bagging[31]. Each classifier in the ensemble will vote on the held-out data, and only precursors where every classifier in the ensemble votes it as a true target are kept in the training data. This filtering resulted in a training set of 2,754,877 peak groups (1,479,571 decoys and 1,275,306 targets) we will refer to as the filtered training data. Linear SVM and non-linear XGBoost models were trained using each training set to create four models (XGB Filter, SVM Filter, XGB No Filter, SVM No Filter). In addition, we trained a PyProphet XGB and a PyProphet LDA model on the unfiltered training data as a comparison to the GPS training method. These six models were applied to evaluate the effects of the training set filtering and model type on model precision, generalization to new data, and the ability to maximize identifications. These models were evaluated on three distinct datasets to confirm they generalize and to ensure that overfitting of the models is not occurring. The yeast test data described above was used as a first evaluation of the models. At a 1% FDR cutoff, PyProphet XGBoost identified less than 1% more precursors per sample compared to GPS XGBoost Filter for the yeast test data (0.45%) (Fig. 2a). The second dataset was 31 mouse-kidney samples searched using a sample-specific spectral library. Here, GPS identified 1.86% more precursors than PyProphet (Fig. 2b). The third dataset was 20 randomly selected undepleted plasma samples from a cohort of 141 patients with septic acute kidney injury (AKI) using a repository scale human tissue spectral library. This AKI test data is used to evaluate the applicability of GPS in large search space scenarios where the proteins in the sample do not closely match the number of proteins in the spectral library. In this case, GPS also identified more precursors than PyProphet (2.58%) (Fig. 2c). The score distributions for all GPS classifiers on each of the test datasets are visualized in Supplementary Fig. S2. In addition, we measured the precision of each GPS classifier on the three test datasets and found GPS XGB Filter to have the highest average precision (0.994) compared to GPS SVM Filter, GPS XGB No Filter, and GPS SVM No Filter (0.987, 0.809, 0.707) (Fig. 2d). Due to the superior number of

identifications passing 1% on the test data and the highest precision score among classifiers, the GPS XGB Filter classifier will be used for the remainder of the study and referred to as GPS. Based on the number of identifications that pass given FDR threshold, these results suggest that GPS is able to generalize more effectively to new data than PyProphet (MK and AKI data) and provide comparable results when scoring data of the exact same sample type (yeast data). The high average precision (0.994) from all datasets, also suggest that GPS is extremely effective at predicting which precursors are correct on a consistent and reproducible basis. The confusion matrices used to calculate precision and predictive performance for each GPS classifier on the three test datasets are available in Supplementary Fig. S1.

**Precursor prediction with GPS enhances identification rates.** Once we demonstrated how GPS could generalize to unrelated data, we then investigated how the high-precision predictions could be used to maximize the identified proteins in an experiment. We analyzed 31 mouse-kidney samples from another study[28] using a spectral library built from 60 data-dependent acquisition (DDA) samples consisting of a mouse-yeast species mixture and compared the number of correct mouse identifications and the entrapment FDR (calculated by counting the number of yeast identifications that pass at a given FDR threshold) obtained using GPS or PyProphet[18]. The effects of peak group predictions are evident in their removal of yeast identifications from consideration and the decrease in the false target portion of the bimodal target distribution of GPS output scores (Fig. 3a, b). In order to boost identification numbers, PyProphet estimates the percent of incorrect targets (PIT or pi0) to downweight decoys when calculating $q$-values to allow more targets to pass at the same FDR[18,32]. Alternatively, using peak group prediction, GPS was able to increase the number of correct mouse identifications while eliminating false yeast identifications, producing lower entrapment FDR across all thresholds compared to PyProphet (Fig. 3d). Over the 31 samples, GPS provides a 50.57% increase compared to PyProphet in the mean number of true mouse precursors that pass 1% FDR control (Fig. 3c).

**Precursor scoring with GPS improves quantitative accuracy.** To establish the quantitative accuracy of identifications produced using GPS, we performed an analysis on a two-species mixture of yeast peptides spiked into a constant mouse-kidney proteome background with two groups of ten technical replicates each. Each group of samples contained the same concentration of Mouse-Kidney proteins, while one group contained 4X more yeast peptides. To verify that our method was validating correct peak groups, we monitored the expected ratios to ensure accurate quantification while still maintaining high levels of identification. In this case, Mouse precursors should expect a 0 log2 fold change, while the yeast precursors should expect a 2 log2 fold change (4X). As a direct comparison, we used OpenSWATH[19] to extract signal followed by scoring with GPS or PyProphet[18]. Scatter plots showing the distributions of log2 fold change between groups of the spike-in data against the mean abundance of the higher-abundance group are visualized in Fig. 4a, b. Horizontal lines in the Fig. 4a, b outline, the area we define as ratio-validated precursors (±0.2 from the expected ratios), and the identifications within these windows are counted at the precursor, peptide and protein level in Fig. 4c. The precursors were considered ratio-validated if their measured log2 fold change fell within the correct window based on the mapped protein label within the ± 0.2 boundaries[15,33]. In comparison to PyProphet, GPS identified 18.97% more precursors, 17.96% more peptides, and 5.28% more proteins in the ratio-validated regions. GPS also decreased the

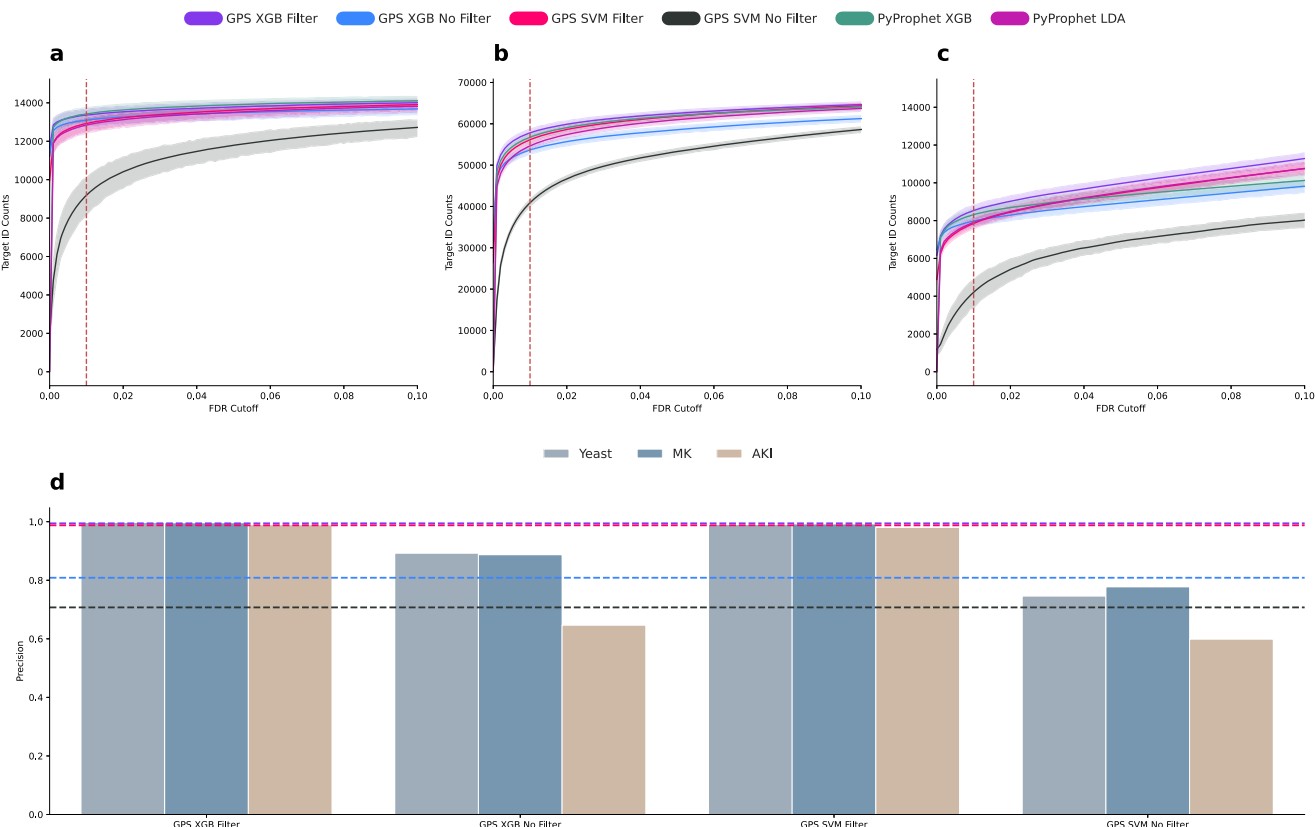

**Fig. 2 Generalization of GPS to three distinctly different sample types.** As a first analysis, we directly evaluated the ability of GPS to generalize to new data. **a–c** Show the average number of precursors identified with the four GPS models (GPS XGB Filter, GPS XGB No Filter, GPS SVM Filter, GPS SVM No Filter) and two PyProphet models (Pyprophet XGB and PyProphet LDA). The dotted red lines represent a 1% FDR cutoff so the performance of each tool on each of the three datasets can be visualized at the specific cutoff. The error bands are based on the 95% confidence intervals calculated at each FDR cutoff. **a** Displays the number of precursors identified on the yeast data, which represents the most simple of the three tested sample types for generalization and the number of proteins is lower, and the number of precursors in the sample directly match the spectral library used. **b** Displays the number of identified precursors for the mouse-kidney data, which represents a more complex proteome. Here, PyProphet does not perform as well as the yeast data, which it was trained on, suggesting that GPS generalizes to new data more effectively. **c** Displays the number of identified precursors at different thresholds for a subset of human plasma samples. These samples were searched using a human tissue library and represent a large search space scenario where the number of precursors does not match the precursors in the spectral library. Here, GPS provides the most identifications at a 1.0% FDR showing how effectively it can generalize independent of the search space. **d** Contains box plots indicating the measured precision for each model at classifying only true targets. The colors of each bar represent the three different datasets. The colors of the horizontal dotted lines correspond to the indicated models in (**a–c**) and are placed at the mean precision for each model across all three datasets. The GPS XGB Filter model had the highest measured average precision across all three sample types. GPS SVM Filter had a comparable measured average precision, indicating the importance of filtering the training data to maximize precision in a precursor classifier.

number of missing values in the overall data by 60.51% (43.73% missing values with PyProphet, and 17.27% missing values with GPS) (Fig. 4d). Based on these metrics, GPS identifies more ratio-validated proteins, while substantially decreasing the number of missing values in the data. To complement the comparison of ratio-validated identifications, we additionally measured the number of identified precursors at increasing thresholds from the expected ratios (Fig. 4e) and the FDR at these same thresholds (Fig. 4f). In Fig. 4e, GPS provides a greater number of identified precursors across all measured thresholds from the expected ratios. In addition, in Fig. 4f, GPS displays a very slight increase in FDR at low thresholds, but a lower overall FDR at the highest measured thresholds. At the ±0.2 threshold, GPS identifies 48680 precursors compared to 40919 with PyProphet (a 18.97% increase) at a 1.90% FDR compared to 1.82% FDR with PyProphet (Visualized by the red dashed lines in Fig. 4e, f). Within the ratio-validated region, GPS identifies almost 8000 more precursors at only a small increase in FDR compared to PyProphet.

**The application of GPS to identify potential protein biomarkers for sepsis-induced AKI.** In order to realize the potential of biomarker discovery experiments using DIA, it can be useful to search blood plasma samples with full human tissue spectral libraries to reach the proteomic depth required to identify interesting and low abundant proteins for further analysis. As an application of GPS, and to evaluate performance in a large search space, we analyzed 141 previously unpublished blood plasma samples from a subcohort of sepsis patients with acute kidney injury from the FINNAKI study[27,29]. These 141 samples are comprised of two established subphenotypes, based on severity of the illness (less severe ($n = 60$) and more severe ($n = 81$)), that were developed from a combination of multiple clinical and molecular markers[30]. We interrogated this data using an optimized human tissue spectral library consisting of the Pan Human Library[5] and appended spectra from direct DIA identifications using MSFragger (v3.5)[34] to correct the retention time and augment the library with more identified precursors (10,952 proteins overall). This analysis puts into context the benefits that GPS

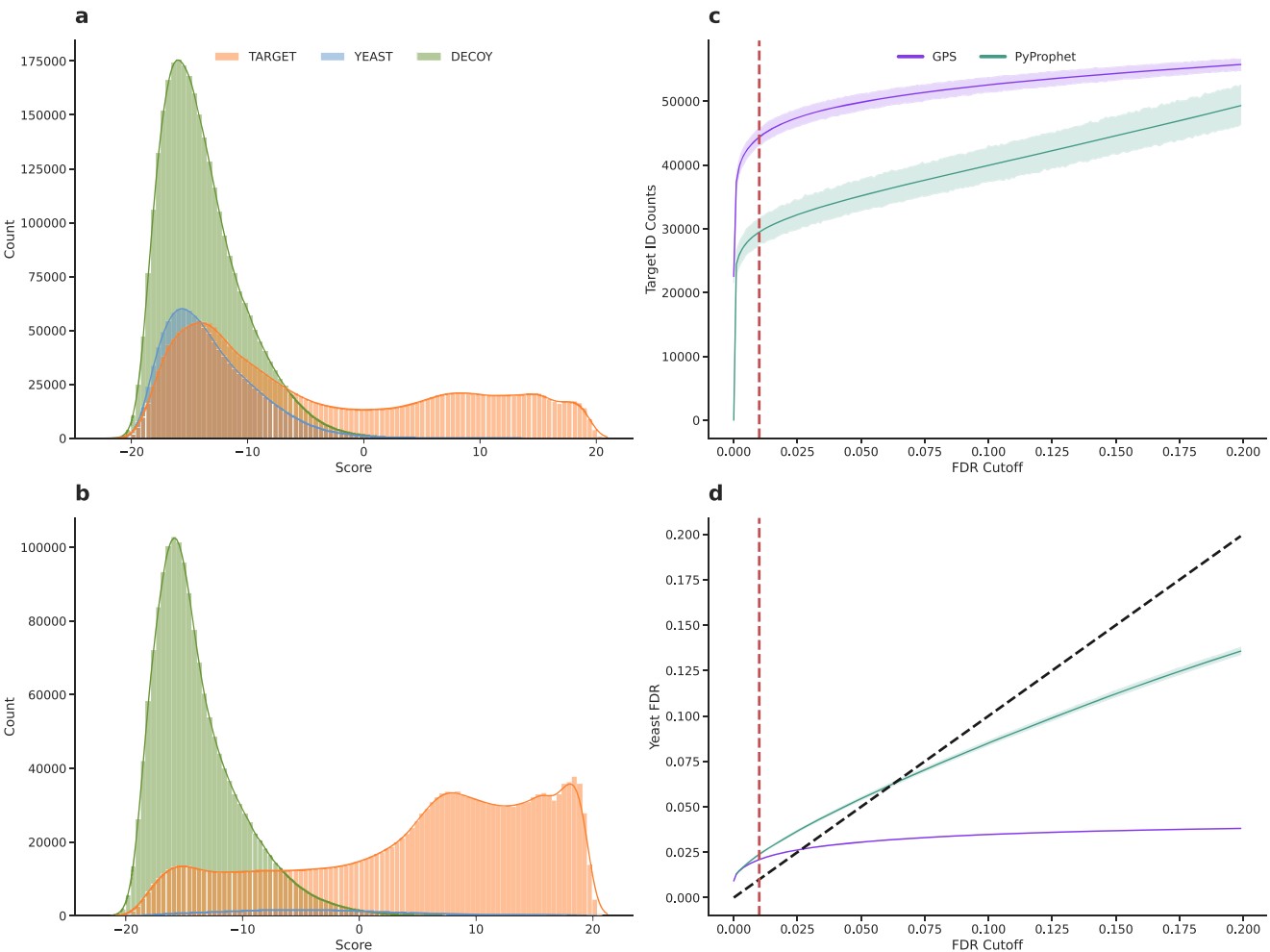

**Fig. 3 Entrapment FDR analysis and precursor identification benchmark.** This figure displays the ability of GPS to eliminate false precursors (Yeast precursors) from analysis using highly precise precursor prediction. On a first pass, precursors are predicted to remove false target precursors from FDR analysis. The precursors that are predicted as true targets are re-extracted to adapt the search space and ensure that only true targets are considered during distribution modeling and FDR calculation. **a, b** Display score distributions calculated by GPS for all extracted precursors in the mouse samples using a mouse-yeast species mixture spectral library. **a** Displays the unfiltered score distributions from GPS for Mouse precursors in the library in orange, Yeast precursors in the library as blue, and Decoys in the library as green. A large peak in the bimodal target distribution can be visualized as overlapping with the yeast distribution and decoy distribution. **b** Displays a filtered score distribution after peak group predictions using GPS and the removal of false targets from consideration. Here we can see that the yeast precursor peak is almost completely eliminated from contention, and the bimodality of the false target (orange target distribution) is lessoned in the region overlapping with the decoy distribution. These two panels display how GPS can control the search space so that the FDR can be controlled in a stable manner. **c** Displays the number of true mouse target precursor counts at increasing FDR thresholds for GPS and PyProphet. The dotted red line indicates a 1% FDR to visualize the performance at that cutoff. The error bands are based on the 95% confidence intervals calculated at each FDR cutoff. At all cutoffs GPS identifies more precursors than PyProphet. **d** Displays the Yeast FDR rates, defined as the number of Yeast identifications divided by the total number of identifications, at increasing FDR thresholds. The red dotted line indicates a 1% FDR and the dotted black line represents $y = x$, where the Yeast FDR should correspond directly to the measured FDR. The error bands are based on the 95% confidence intervals calculated at each FDR cutoff. The measured Yeast FDR is lower using GPS at all thresholds compared to PyProphet, and is more strict at higher thresholds than PyProphet while still identifying more precursors at the same thresholds.

provides when querying a large search space and the benefit of using extensive curated repository spectral libraries in discovery DIA. We first compared the standard OpenSWATH/PyProphet workflow with GPS to see the advantages provided when interrogating a larger search space. In order to get PyProphet to run without failure on this data, it was necessary to remove the pi0 estimation[32] as this failed due to the large number of false target identifications in the spectral library. Here, the estimated number of false targets was greater than the number of decoys in the analysis, so pi0 estimation must be disabled or manually set to enable PyProphet to run through without error. Since the goal of this analysis was to try to identify biological differences between AKI subphenotypes, we focused on the number of comparable

proteins per group as a proxy for method performance. For this metric, we counted the number of proteins quantified in at least ten biological replicates per group and found GPS identified 24.35% more than PyProphet (771 vs. 620). Finally, we compared the number of statistically significant differentially abundant proteins (at least ten replicates per group) with a corrected $P$ value < 0.1 and found a 22.91% increase for GPS (338 vs. 275). Volcano plots for both PyProphet (Fig. 5a) and GPS (Fig. 5b) are visualized as well as the overall protein counts (Fig. 5c). Overall, GPS quantified 1312 proteins, 53.81% more than PyProphet. Using the increased analytical depth of GPS, we applied recursive feature elimination (RFE) with cross-validation via explainable artificial intelligence (SHAP[35]) and an XGBoost classifier to

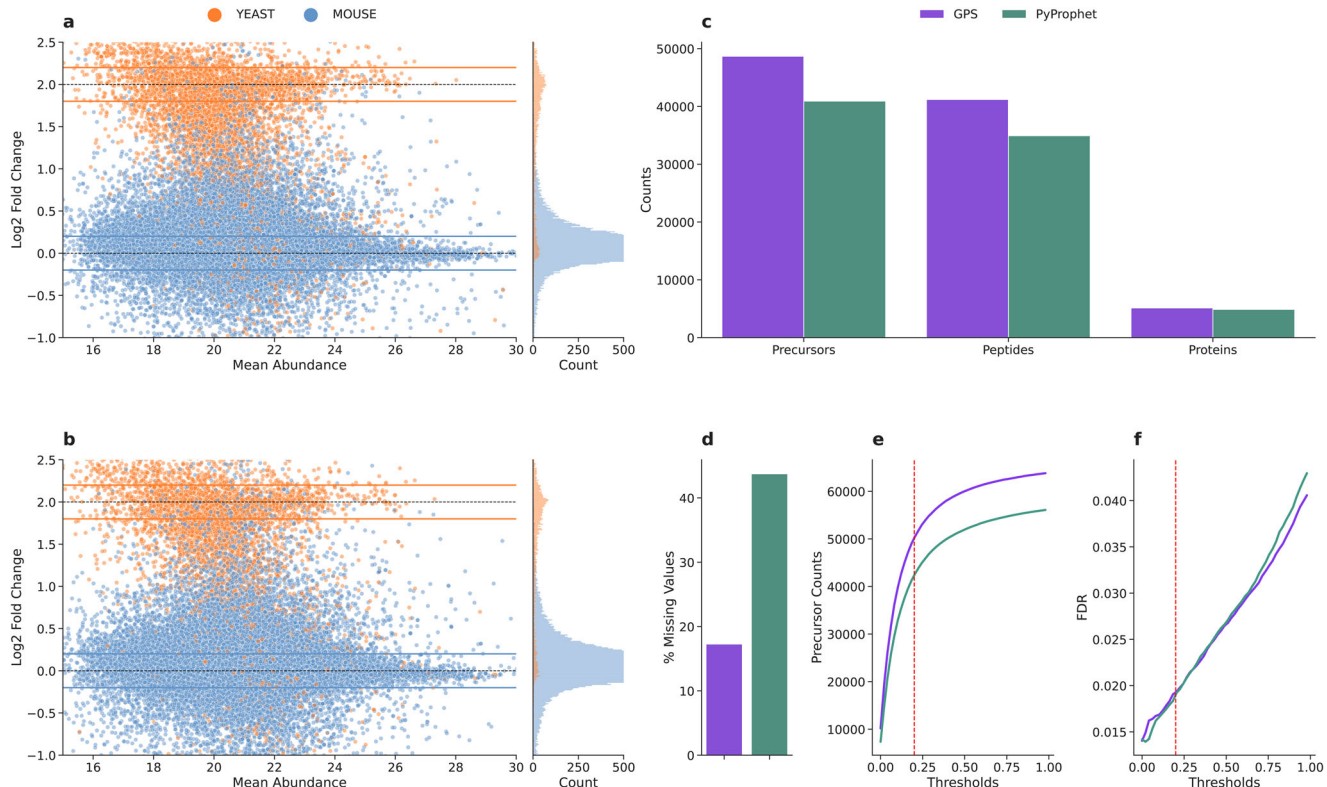

**Fig. 4 Quantification accuracy of GPS evaluated by a two-species mixture spike-in dataset.** We evaluated the quantification accuracy of GPS by analyzing a two-species mixture of yeast peptides spiked-in into a constant mouse-kidney proteome background with two groups of ten technical replicates each. Each group of samples contained the same concentration of Mouse-Kidney proteins, while one group contained 4× more yeast peptides and we measured the number of precursors that mapped correctly into the expected ratio of their species (0.0 ± 0.2 log2 fold change for Mouse precursors and 2.0 ± 0.2 log2 fold change for Yeast precursors. **a** displays the mean abundance of precursors identified using GPS against their log2 fold change and colored by their mapped species. Histogram plots directly to the right of these scatter plots display the distribution of the species mixture on the log2 fold change scale. The expected ratio regions are highlighted to display which precursors were considered as ratio-validated. **b** displays the same as (**a**) but for PyProphet. **c** Displays the overall counts of ratio-validated precursors, peptides, and proteins, from the regions highlighted in (**a**, **b**) for GPS and PyProphet. From these validated regions, GPS identifies more precursors, peptides, and proteins than PyProphet. **d** Shows the percentage of missingness in the quantitative matrices for GPS and PyProphet. Here, GPS decreased the number of missing values by 60.51% compared to PyProphet. This is important in context with (**c**), as GPS is able to provide a greater number of accurately quantified precursors and a substantially more complete data matrix as measured by the % missing values. In order to provide an evaluation beyond the ratio-validated cutoff, we measured the number of identified precursors and the FDR at increasing log2 fold change thresholds from the expected ratios of the species mixture in (**e**, **f**). **e** Displays the number of precursors identified and quantified at increasing thresholds from the expected values. GPS identifies more precursors at every threshold compared to PyProphet. **f** Displays the the FDR as a function of increasing thresholds from the expected ratios of each proteome in the mixture. Here, we can see at low thresholds, GPS displays a slightly higher FDR, but the two tools even out over the measured thresholds, with GPS having a lower FDR further away from the expected ratios. GPS is able to identify more precursors while maintaining a comparable FDR to PyProphet over the increasing thresholds measured. The dotted horizontal lines visualize the number of precursors and measured FDR at the ±0.2 thresholds used for ratio-validated quantification.

identify a panel of highly accurate predictive proteins in differentiating between AKI subphenotypes. SHAP-RFE analysis identified a group of 18 proteins as the most effective discriminatory panel (Fig. 5d), with the CD44 antigen (CD44) as the overall most important in classification. We evaluated the performance of this classifier using cross-validation (tenfold) and calculated an accuracy of 0.86 with a standard deviation of 0.11. In addition, these proteins were able to accurately cluster AKI subphenotypes in an unsupervised manner (Rand score 0.82), further confirming their effectiveness in stratifying the two AKI subphenotypes (Fig. 5e). The abundance profiles between groups can be visualized in Fig. 5f, and their average fold change and statistical significance can be visualized with green in Fig. 5b. A few noteworthy proteins (Catalase (CATA), Clusterin (CLUS), Collagen alpha-2(I) chain (CO1A2)) are not considered statistically significant, but are still considered important in the context of differentiating between subphenotypes. The protein Cathepsin Z (CATZ) from the panel was not considered for DE analysis as it

was not quantified in the minimum number of replicates for the less severe subphenotype but consistently quantified in the more severe.

## Discussion
With the implementation of our new methods, we showed how GPS can effectively generalize to new data, increase the number of identifications, boost quantification accuracy, and be applied in a discovery DIA analysis to investigate a complex biological question. We were able to demonstrate how these established methods can suffer when the spectral library search space is too large and does not match the sample, while GPS is able to score data in this scenario in a stable manner while increasing the number of identifications and quantitative accuracy. These combined improvements allow for the deep and in-depth analysis of plasma proteome samples using repository scale spectral libraries to boost the power of discovery DIA experiments using a completely open-source tool chain.

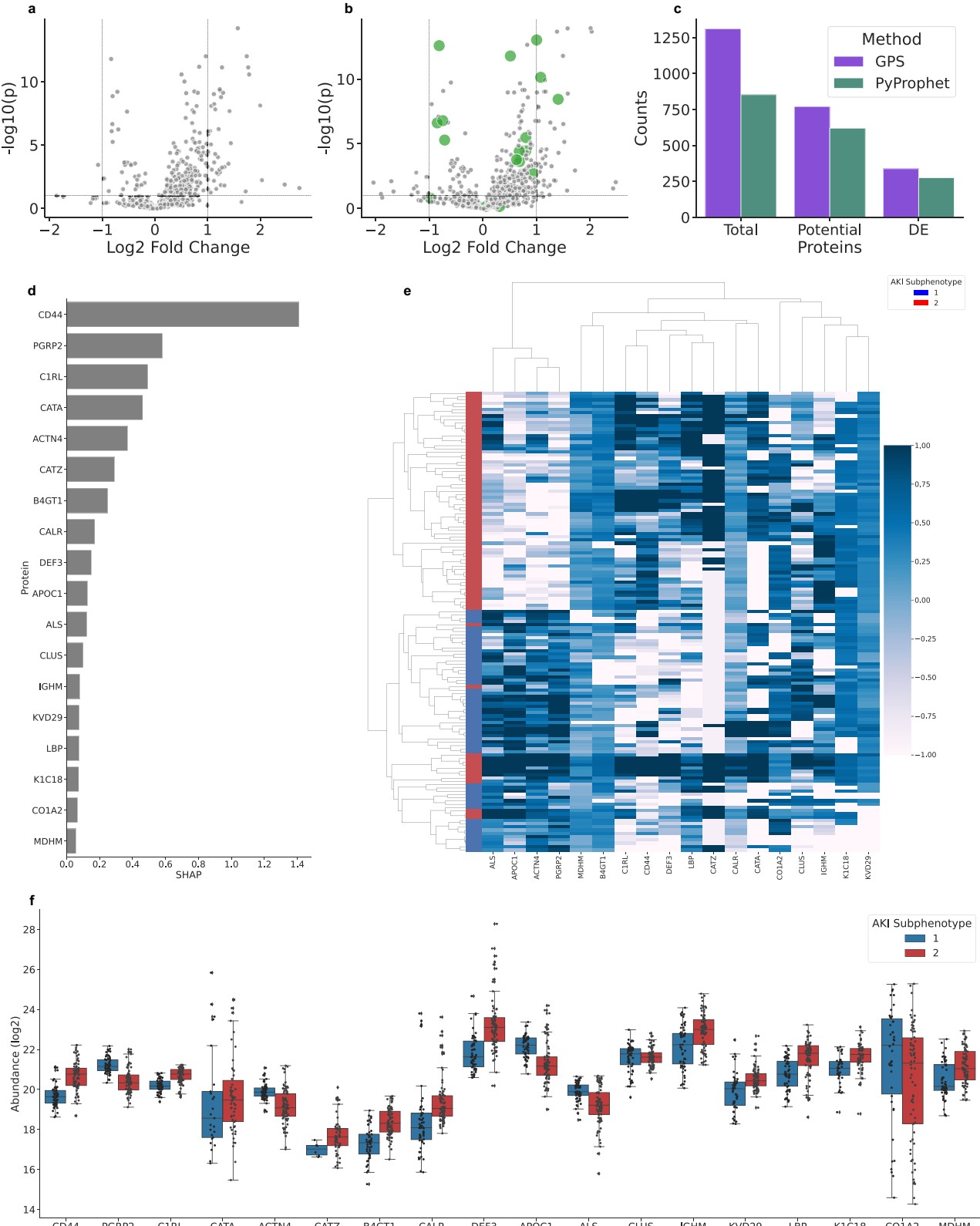

Our denoising method filters training data effectively to ensure only true precursors are included when building the model without overfitting on experiment-specific data. In our model evaluation analysis, the performance of PyProphet decreased as it was applied to new unrelated data while GPS remained consistent suggesting that PyProphet may have slightly overfitted to the training data. These performance improvements suggest that the GPS training method produces classifiers that generalize more effectively to diverse experiment types than the semi-supervised approach used with PyProphet. In addition, there is no need to optimize hyperparameters to squeeze the best performance out of GPS, as the generalized model will predict and score new precursors accurately in a stable manner no matter the conditions of the data. PyProphet can be optimized to train sample-specific

**Fig. 5 The application of GPS for the identification of blood-based biomarkers in septic AKI.** The analysis performed in this application serves two main purposes. One, to evaluate GPS in a large search space and compare the number of potentially comparable proteins to PyProphet. Two, to apply GPS to identify a group of biomarkers using machine learning with recursive feature elimination and explainable artificial intelligence (RFE-SHAP) that could be useful in stratifying subphenotypes of septic AKI (total $n = 141$, less severe ($n = 60$), and more severe ($n = 80$)). **a** Displays a Volcano plot for differentially abundant proteins identified using PyProphet. **b** Displays a Volcano plot for differentially abundant proteins identified using GPS with the 18 proteins selected as potential biomarkers highlighted in green. **c** Displays the overall counts of the total proteins identified by each method (GPS and PyProphet), the potential proteins (proteins found in minimum ten replicates per group), and the statistically significant differentially abundant proteins (corrected $P$ value < 0.1). At all levels, GPS identified more proteins than PyProphet for the measured data. To identify a group of proteins that could be important in differentiating between subphenotypes of septic AKI, we employed machine learning and RFE-SHAP to pick the optimal set of proteins used for classification. **d** Displays the 18 proteins selected using RFE-SHAP analysis and their mean importance calculated by SHAP in predicting AKI subphenotypes. CD14 was found as largely the most important protein, with many other documented infection and inflammation markers included in the list. **e** Shows a clustermap of the AKI samples using the 18 selected proteins. Colored by subphenotype on the y-axis, it is clear that the selected proteins are accurate in stratifying the defined AKI subphenotypes. **f** Visuzlies the box and swarm plots for the abundance of the 18 selected proteins grouped by AKI subphenotype. The boxes represent the interquartile range of the protein abundances with the swarm plot showing the individual measurements.

classifiers in different conditions, but a meticulous process to find optimal hyperparameters is required. This nontrivial and time-consuming parameter optimization and model training can be avoided completely by applying generalized models, such as GPS, directly to new data as they have already been trained and evaluated.

In standard proteomics experiments, the estimation of the percent of incorrect targets (PIT) has become an established method to boost the number of identifications that pass through at a given FDR cutoff in mass-spectrometry proteomics[18,32] However, predicting this percentage of false targets is computationally difficult, as it is unknown which targets are in fact false, or where to split the target distribution to estimate the PIT (pi0). Existing methods use certain heuristics to estimate pi0 by calculating the difference between the decoy counts and target counts at certain score cutoffs[32], or provide naive counts based on the number of targets below a 1.0% FDR, but this can lead to inaccurate results depending on the shapes of the score distributions. In our analysis, we see that PyProphet, which by default uses PIT estimation when calculating $q$-values, leads to an increase in the number of decoys (and therefor false targets) that pass at a given score threshold. Instead of using this method to boost identifications, GPS can increase IDs and decrease the number of false hits that pass FDR control by first predicting true target peak groups and using those predictions to control the FDR, which provides a distinct advantage over typical PIT estimation while providing the same benefits.

The semi-supervised algorithm with pi0 estimation in PyProphet can provide great results on individual experiments, particularly by increasing the number of identifications, as it maximizes the local number of targets that are validated. However, this could potentially introduce false positives that are elucidated only when analyzing the quantitative accuracy of known spiked-in proteins. We demonstrate here that the optimized GPS prediction model outperforms PyProphet based on the number of ratio-validated identifications in the spike-in data, showing that a pre-trained model can be used to increase the accuracy of quantification without training new models that do not generalize well to new data. Ideally, we would have liked to use the subscores calculated by DIA-NN to train a GPS classifier and then apply the GPS framework within the DIA-NN tool chain, but since DIA-NN is closed source and does not expose or record the subscores calculated to estimate precursor quality, this direct comparison and integration was not possible. Given these improvements, we were still able to bring GPS close to the performance of DIA-NN on the spike-in data in terms of overall identifications, and even surpass DIA-NN when it comes to decreasing the number of missing values and higher quantitative accuracy closer to the expected ratios of the spike-in data (Supplementary Fig. S3). We

believe that if DIA-NN implemented a strategy where a static model was used to predict and score precursors, that some of the same benefits we see in the OpenSWATH pipeline could be realized to further improve identification and quantification with DIA-NN.

In theory, it is particularly beneficial to search blood plasma samples with full human tissue spectral libraries to delve deeper into the proteome to identify potential disease markers missed in standard analysis, especially low abundant proteins. These large libraries create a search space imbalance where the true target labels are extremely noisy, i.e., <5% of the targets extracted by the library are true targets. Semi-supervised methods attempt to eliminate noise in an iterative fashion using an algorithm where new targets are selected based on $q$-value cutoffs[17,18], but when the true target-to-decoy ratio is small the selected targets may be especially noisy, leading to a class imbalance. There are different ways to try and correct training set imbalances, such as downsampling the majority class or upsampling the minority class[26], and it is possible to provide the class ratios to certain machine-learning algorithms to ensure that over-represented classes do not dominate the training loops. In the case of GPS, instead of choosing to implement imbalanced learning methods with sample-specific classifiers, we trained models on curated data that generalize to unrelated samples, so that it does not matter if there is a class imbalance or noisy labels in the data being predicted and score. To ensure that the no class imbalance issues effect training, we calculate the class ratios and pass them to our training algorithms so that weights are adjusted accordingly and the imbalance is taken into account during training.

Taking the described algorithmic benefits into consideration, we applied GPS to analyze undepleted plasma AKI samples as an example of a biological application. From the 1312 quantified proteins, RFE-SHAP analysis was able to identify a panel of 18 with high accuracy and separating power that indicates they would provide a good starting point for investigation as potential protein biomarkers. In fact, many of these proteins have already been identified and studied as potential sepsis markers or markers for infection and inflammation[36–45]. The majority of the 18 proteins were higher in abundance in the more severe subphenotype, such as CD44 antigen (CD44), Complement C1r subcomponent-like protein (C1RL), Beta-1,4-galactosyltransferase 1 (B4GT1), and Lipopolysaccharide-binding protein (LBP). Proteins that were lower abundant in the more severe subphenotype were Insulin-like growth factor-binding protein complex acid labile (ALS), Apolipoprotein C-I (APOC1), Alpha-actinin-4 (ACTN4), and N-acetylmuramoyl-L-alanine amidase (PGRP2). This panel could further be expanded to any protein that has a significant weight in classifying the severity of AKI based on a combination of SHAP values and differential

expression analysis in an effort to identify novel disease bio-markers. These findings suggest that it is possible to identify potential sepsis markers in plasma samples and accurately quantify them using repository scale spectral libraries and precursor prediction with GPS. These added benefits could significantly aid in the stratification of sepsis subphenotypes by allowing for a deeper exploratory investigation of the plasma proteome on a systematic basis and the informed data-driven selection of potential biomarkers for further validation. This approach would also generalize to other biological conditions or diseases easily, providing a systematic method towards discovery DIA with GPS.

Overall, we have proposed GPS as a method for the statistical validation of DIA mass-spectrometry data and provided evidence that generalized scoring models can outperform dynamically trained models especially in a large search space environment by utilizing precursor prediction for stable FDR control for downstream quantification. Further, we provide evidence that sophisticated generalized scoring models can be used in tandem with massive-scale spectral libraries to support the development of discovery proteomics in DIA mass spectrometry.

## Methods

**Datasets**. An overview of all data used in the study, along with associated metadata, is completely summarized in Table 1.

In order to provide a chromatographically diverse set of training data, we used a dataset comprised of 129 different samples of 500ng Yeast tryptic digest (Promega) with varying gradient lengths (30, 45, 60, 90, 120 min) and acquired with DIA. This dataset will be referred to as the yeast dataset. The mass-spectrometry proteomics data have been deposited to the ProteomeXchange Consortium via the PRIDE[46] partner repository with the dataset identifier PXD038367.

We generated a spike-in dataset of known concentrations of Yeast tryptic digest peptides (Promega) spiked into a constant mouse-kidney background. The mouse-kidney material was obtained from the previous project[28]. All animal use and procedures were approved by the local Malmö/Lund Institutional Animal Care and Use Committee, ethical permit number 03681-2019. Animals were treated in accordance with the National Institutes of Health for the Care and Use for Laboratory Animals. Nine-week-old Female C57BL/6 mice (Janvier, Le Genest-Saint-Isle, France) were sacrificed, and kidneys were isolated into a tube containing DPBS and silica beads (1 mm diameter, Techtum). The kidneys were then homogenized using MagNAlyser (Roche) and stored at −80 °C. Homogenates were then thawed and centrifuged at 10,000 × g for 10' at 4 °C. The supernatant containing the soluble proteins was collected, and protein content was estimated using BCA (Pierce). In total, 25 μg of protein was taken for reduction, alkylation, digestion, and C18 clean-up, as described below. A serial dilution series of yeast peptides (1×, 2×, 4×, 8×, 16×, 32×) was performed, and ten technical replicates of each concentration were sampled for a total of 60 samples. Internal retention time peptides were also added to each sample. Each of the 60 samples were analyzed on the Orbitrap HF-X using both DDA and DIA. The mass-spectrometry proteomics data have been deposited to the ProteomeXchange Consortium via the PRIDE[46] partner repository with the dataset identifier PXD038377.

In total, 31 mouse-kidney samples were selected from a previous study[28] to provide a base for the entrapment FDR analysis. The samples are from the same study as the mouse-kidney material used to prepare the spike-in data, and follow the same ethical considerations and approvals (ethical permit number 03681-2019), as well as sample preparation methods described above.

AKI plasma samples used in the study belong to the FINNAKI study[27], a prospective, observational, multicenter study evaluating the development of AKI in ICU patients with sepsis and septic shock, in accordance with the Helsinki Declaration. The Ethics Committee of the Department of Surgery, Helsinki and Uusimaa Hospital District, approved the study protocol, and each participant or

their proxy gave written informed consent. The Ethics Committee of the Department of Surgery, Helsinki and Uusimaa Hospital District, also approved the inclusion of participants for all centers involved as well as the use of deferred consent (Reference Number 18/13/03/02/2010). Patient demographics, medical history, severity scores, length of stay, physiologic data, and hospital mortality were collected from the Finnish Intensive Care Consortium prospective database (Tieto Ltd, Helsinki, Finland) with a study-specific case report form. AKI status was screened at admission and during the first 5 days of ICU stay. All data collection was blinded to the index test results. Plasma samples were collected immediately at ICU admission or after 2 h at the latest and directly centrifuged, aliquoted, and frozen to −80 °C. Samples were sent on dry ice from Helsinki, Finland, to Lund, Sweden, for mass-spectrometry analysis. AKI was defined according to the Kidney Disease: Improving Global Outcomes (KDIGO) criteria based on changes in serum creatinine[47]. In total, 51% of the patients developed AKI within the first 5 days in the ICU, with 30% diagnosed <12 h from admission. Approximately 100 patients each developed stage 1, 2, and 3 AKI. 91 patients received RRT, and the 90-day mortality for AKI patients was 33.7%. Overall, 141 samples were chosen for up to 5 time points from 23 acute kidney injury patients. The patients were from two distinct subphenotypes that were previously defined using a panel of clinical markers and latent class analysis[30]. No power analysis was performed, the 23 patients were selected on the basis of culture-positive sepsis. The mass-spectrometry proteomics data have been deposited to the ProteomeXchange Consortium via the PRIDE[46] partner repository with the dataset identifier PXD038394.

**Mass-spectrometry sample preparation and data acquisition**. All sample preparation steps of the 141 AKI samples, including desalting and protein digestion, used the Agilent AssayMAP Bravo Platform (Agilent Technologies, Inc.) per the manufacturer's protocol. Each plasma sample was diluted 1:10 (100-mM ammonium bicarbonate (AmBic); Sigma-Aldrich Co, St Louis, MO, USA), and 10 L of each diluted plasma sample were transferred to a 96-well plate (Greiner G650201) where 40 μL of 4 M urea (Sigma-Aldrich) in 100 mM AmBic was manually added with a pipette for a final volume of 50 μL. The proteins were reduced with 10 μL of 60 mM dithiothreitol (DTT, final concentration of 10 mM, Sigma-Aldrich) for 1 h at 37 °C followed by alkylation with 20 μL of 80 mM iodoacetamide (IAA, final concentration of 20 mM, Sigma-Aldrich) for 30 min in the dark at room temperature. The plasma samples were digested with 2 μg Lys-C (FUJIFILM Wako Chemicals U.S.A. Corporation) for five hours at room temperature and further digested with 2 μg trypsin (Sequencing Grade Modified, Promega, Madison, WI, USA) overnight at room temperature[48]. The digestion was stopped by pipetting 20 μL of 10% trifluoroacetic acid (TFA, Sigma-Aldrich), and the digested peptides were desalted on Bravo platform. To prime and equilibrate the AssayMAP C18 cartridges (Agilent, PN: 5190-6532), 90% acetonitrile (ACN, Sigma-Aldrich) with 0.1% TFA and 0.1% TFA were used, respectively. The samples were loaded into the cartridges at the flow rate of 5 μL/min. The cartridges were washed with 0.1% TFA before the peptides were eluted with 80% ACN/0.1% TFA. The eluted peptides were dried in a SpeedVac (Concentrator plus Eppendorf) and resuspended in 25 μL of 2% ACN/0.1% TFA. The peptide concentration was measured using the Pierce Quantitative Colorimetric Peptide Assay (Thermo Fisher Scientific, Rockford, IL, USA). The samples, 10 μL, were diluted with 10 μL 2% ACN/0.1% TFA and spiked with synthetic iRT peptides (JPT Peptide Technologies, GmbH, Berlin, Germany) before liquid chromatography-mass-spectrometry (LC-MS/MS) analysis.

All additional protein samples (Yeast and spike-in samples) were denatured with 8 M urea and reduced with 5 mM Tris(2-carboxyethyl)phosphine hydrochloride, pH 7.0 for 45 min at 37 °C, and alkylated with 25 mM iodoacetamide (Sigma) for 30 min followed by dilution with 100 mM ammonium bicarbonate to a final urea concentration below 1.5 M. Proteins were digested by incubation with trypsin (1/100, w/w, Sequencing Grade Modified Trypsin, Porcine; Promega) for at least 9 h at 37 °C. Digestion was stopped using 5% trifluoroacetic acid (Sigma) to pH 2–3. The peptides were cleaned up by C18 reversed-phase spin columns as per the manufacturer's instructions (Silica C18 300 Å Columns; Harvard Apparatus). Solvents were removed using a vacuum concentrator (Genevac, miVac) and were resuspended in 50 μl HPLC-water (Fisher Chemical) with 2% acetonitrile and 0.2% formic acid (Sigma).

Peptide analyses were performed on a Q Exactive HF-X mass spectrometer (Thermo Fisher Scientific) connected to an EASY-nLC 1200 ultra-HPLC system

**Table 1 An overview of the data used in the study and their uses.**

| Dataset | Acquisition | Samples | Replicates | Usage | PRIDE ID |
|---|---|---|---|---|---|
| Yeast | DIA | 129 | All | Generalizable model training | PXD038367 |
| Mouse-Yeast | DDA | 60 | 10 | Spectral library generation | PXD038377 |
| Mouse-Yeast | DIA | 60 | 10 | GPS Benchmark | PXD038377 |
| Mouse Kidney | DIA | 31 | None | Entrapment FDR Analysis | |
| AKI | DIA | 141 | None | Large Search Space Comparison Optimized Library Analysis ML Differential Expression | PXD038394 |

(Thermo Fisher Scientific). Peptides were trapped on precolumn (PepMap100 C18 3 µl; 75 µl × 2 cm; Thermo Fisher Scientific) and separated on an EASY-Spray column (ES903, column temperature 45 °C; Thermo Fisher Scientific). Equilibrations of columns and sample loading were performed per the manufacturer's guidelines. Mobile phases of solvent A (0.1% formic acid), and solvent B (0.1% formic acid, 80% acetonitrile) was used to run a linear gradient from 5 to 38% over various gradient length times at a flow rate of 350 nl/min. The 44 variable windows DIA acquisition method is described in ref. [49]. MS raw data were stored and managed by openBIS (20.10.0)[50] and converted to centroided indexed mzML files with ThermoRawFileParser (1.3.1)[51].

**Spectral library creation**. An experiment-specific library for the spike-in data was built by analyzing the samples acquired using DDA using FragPipe (v18.0). First the samples were searched using MSFragger (v3.5)[52] with default parameters using a FASTA file of Swiss-Prot reviewed *Saccharomyces cerevisiae* and *Mus musculus* proteomes concatenated with reverse sequence decoy proteins. Peptide spectrum matches (PSMs) were validated using Percolator[17]. The Philosopher toolkit (v4.4.0) was used to perform protein level FDR control with ProteinProphet, generate downstream reports, and filter the resulting identifications[53]. The Python package easypqp was then used to convert and format the library for use by OpenSWATH, and the OpenMS (3.0.0) tool chain was used to create decoys using the Open-SwathDecoyGenerator command with default settings. 10 spiked-in retention time peptides (iRT) were added for initial alignment and retention time correction for each sample.

To augment the PHL[5] with additional identifications and correct the retention time to the experiment at hand, we first searched the 141 AKI plasma samples using MSFragger-DIA (v3.5)[34]. Using the resulting set of identifications, shared precursors between the PHL and direct search were selected for retention time alignment. LOWESS was first used to smooth the correlation between the direct search results and the PHL and then an interpolated univariate spline function was fit on top of this to adjust the retention time in the direct search to the scale of the PHL. The shared proteins between the two libraries were replaced in the PHL with proteins, and associated precursors, from the direct search, and the proteins not contained in the PHL were appended to the library. OpenSWATH and the OpenMS (3.0.0) toolchain was used to create decoys using the OpenSwathDecoyGenerator command with default settings. Ten spiked-in retention time peptides (iRT) were used for initial alignment and retention time correction for each sample.

**GPS**. GPS is a Python library and command line utility for the generalized statistical validation of precursors. The source can be found here (https://github.com/InfectionMedicineProteomics/gps). GPS leverages the package numpy[54] for efficient processing of numerical data, scikit-learn, sklearn and xgboost for implementing machine-learning algorithms, and numba[55] for its just-in-time (JIT) compilation that compiles Python to machine code for optimization in performance-critical areas of the library.

The model details are available at https://github.com/InfectionMedicineProteomics/gps. Input features for the training of the models are based on MS2 level subscores as calculated by OpenSWATH[19] and are as follows: var_bseries_score, var_dotprod_score, var_intensity_score, var_isotope_correlation_score, var_isotope_overlap_score, var_library_corr, var_library_dotprod, var_library_manhattan, var_library_rmsd, var_library_rootmeansquare, var_library_sangle, var_log_sn_score, var_manhattan_score, var_massdev_score, var_massdev_score_weighted, var_mi_score, var_mi_weighted_score, var_norm_rt_score, var_xcorr_coelution, var_xcorr_coelution_weighted, var_xcorr_shape_weighted, var_yseries_score.

The filter and no filter SVM models used in the study were trained using stochastic gradient descent (SGD)[56] using sklearn. We used hinge as the loss function with l2 regularization, an alpha of 1e-5 with an adaptive learning rate and early stopping. Class imbalance ratios were passed to the training function to properly weight each sample. The implementation details are contained in the source code on github.

The filter and no filter XGB models used in the study were trained using logloss as the evaluation metric and "logitraw" as the objective. Class imbalance ratios were passed to the training function to properly weight each sample. The implementation details are contained in the source code on github. The XGB Filter model is used through the study and is referred to as GPS.

The denoising algorithm used to filter the Yeast training set is based on the concept of bagging from machine learning[31]. The data to be analyzed is first split into k number of folds (default is 10, and what is used throughout the study). Each fold is scored by training an ensemble of n logistic regression classifiers (default is 10, and what is used throughout the study) using stochastic gradient descent[56] on data that is randomly sampled with replacement from the data left out of the selected k-fold. The ensemble of classifiers is then used to score the k-fold data, providing an average target probability for each precursor in the fold, and voting on each precursor to determine the vote percentage. A vote is considered a positive vote if the predicted probability for the individual classifier in the ensemble exceeds a threshold.

To remove the noisy labels from the training dataset, the denoising algorithm described above was used to calculate a vote percentage for each precursor. If the calculated vote percentage was 100% then the precursor was kept as a true target.

The probability to accept a positive vote was set at 0.75 to more strictly filter out potential false positives at the risk of losing some true identifications in the dataset. The negative training set, the decoy precursor, remained unfiltered.

The algorithm used to score each precursor in GPS is very straightforward. The precursor and their associated subscores are read in and parsed into a data structure that exposes the selected subscores for each peak group. For prediction, the chosen GPS model is used to predict whether or not a precursor is a true precursor or not, and the results along with the calculated GPS score (DScore) is written to an output file. The same procedure is done when the GPS is used to score a sample, except inference via prediction is not performed and only the DScore is calculated for a precursor.

Q-values are calculated using an implementation of the qvality algorithm[57], where an interpolated spline is fit to the distributions of the target and decoy scores. A q-value for a particular precursor is calculated by first integrating the area under the curve of the decoy distribution from that particular score to the max to get the decoy counts at a particular score threshold. The target counts are then obtained by integrating the area under the curve of the target distribution from that particular score to the max. Finally, the decoy counts are divided by the target counts plus the decoy counts to calculate a q-value, which can be used to filter for a given FDR. The highest-scoring precursor for each precursor and the corresponding scores and q-value are written out to a file for downstream processing. In addition, a more basic and faster q-value calculation method has been implemented using decoy counting and is available from the CLI and Python library API of GPS.

Global FDR control is implemented in a similar manner to PyProphet[18], where all scored samples in an experiment are aggregated and the highest-scoring precursor is selected to represent either the peptide or the protein at the desired level. Once the highest-scoring precursors are selected, q-values are estimated using the method described above. The resulting scoring models are exported as serialized Python objects that can then easily be used from the command line by GPS to export an annotated quantitative matrix.

GPS can aggregate all scored samples, and the global peptide and protein models, into a quantitative matrix for downstream analysis. Each sample is read into a data structure that filters the samples in the precursor based on their individual q-values. Once all samples have been parsed, they are annotated with their global peptide and protein level q-values using the score distribution objects that were previously built. The resulting annotated quantitative matrix is then written out for downstream analysis by the tool of your choosing.

For the overall workflow, we adapted the previously published DIAnRT workflow[58] to optimize signal extraction at the sample level before combining the analysis to control for the global FDR. To do this, a first iteration is performed where sub-optimal retention time peptides are provided to align a sample to the spectral library. GPS is used to then predict which extracted precursors are true precursors, and then the highest-scoring precursors from a specified number of bins are selected and written out to a sample-specific retention time library. The precursor predictions are aggregated across all samples and combined into a second-pass spectral library where the sample-specific retention time libraries are used to align and correct the retention time and mass-to-charge ratios to the spectral library with more stringent parameters in a second pass which is scored using GPS. These final validated precursors are then used to calculate the peptide and protein level FDR using the approach implemented in PyProphet[18] to produce a quantitative matrix. Software to perform the sample-specific retention time library extraction can be found in combination with the GPS python package and complete snakemake workflows and corresponding command line options for the different tools used can be found at (https://github.com/InfectionMedicineProteomics/GPSWorkflows).

**Statistics and reproducibility**. All downstream analysis was performed using the Data Processing Kitchen Sink (DPKS) Python package for general-purpose data processing of mass-spectrometry proteomics data (https://github.com/InfectionMedicineProteomics/DPKS).

For all datasets, a retention time-mean sliding window normalization method was used based on the implementation in the NormalyzerDE R package[59]. Proteins were quantified for the AKI analysis using an implementation of the iq relative quantification algorithm[60]. Differential expression was performed using linear models, at the precursor level for the spike-in analysis and protein level for the AKI analysis. Multiple testing correction was performed using the Benjamini–Hochberg method[61]. All of these methods, including other options, are available in the DPKS package.

In order to provide context and a ranking to the differentially expressed proteins, we trained an XGBoost classifier using quantified proteins from GPS and DPKS to classify between the subphenotypes in the AKI analysis. Missing values in the quantitative matrix were first imputed with zero values, as it is assumed if the protein was not quantified and identified that it is not in the sample. The protein quantities are then scaled to remove the mean and scale to unit variance. The model was evaluated using tenfold cross-validation to provide mean accuracy. We used the SHAP[35] python package to then calculate the relative importance of each protein in differentiating between the subphenotypes of AKI and recursive feature elimination as implemented in sklearn. It was then possible to rank the differentially expressed proteins by their relative importance instead of setting arbitrary P value and log2FC cutoffs to identify proteins and select a panel of 18 potential protein biomarkers.

**Reporting summary**. Further information on research design is available in the Nature Portfolio Reporting Summary linked to this article.

## Data availability

The mass-spectrometry proteomics data have been deposited to the ProteomeXchange Consortium via the PRIDE[46] partner repository with the dataset identifiers PXD038367, PXD038377, and PXD038394 for the Yeast, Mouse-Kideny, and AKI data respectively. All results from the analysis are available in Supplementary Data 1. The GPS models trained and used throughout the study, together with their associated scalers, are available in Supplementary Data 2.

## Code availability

All GPS code is open-source and freely available under the MIT license at https://github.com/InfectionMedicineProteomics/gps. All DPKS code for downstream analysis is open-source and freely available under the MIT license at https://github.com/InfectionMedicineProteomics/DPKS.

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

## Acknowledgements

L.M. was supported by the Swedish research council (grant number VR-2020-02419), the Wallenberg Foundation (grant number 2016.0023), and Alfred Österlunds Foundation. J.M. was supported by the Wallenberg Foundation (WAF grant number 2017.0271), the Swedish research council (grant number 2019-01646 and 2018-05795), and Alfred Österlunds Foundation.

## Author contributions

A.S. and L.M. conceptualized GPS. A.S. provided the implementation of GPS and the DPKS software packages, the analysis and experimental design, and wrote the manuscript. C.K. contributed to the development and mass-spectrometry analysis of the Yeast, Mouse-Kideny, and spike-in data, and to the writing of the manuscript. T.M. contributed to the development and mass-spectrometry analysis of the Mouse-Kidney and AKI data, and to the writing of the manuscript. E.H. contributed to the SHAP-RFE AKI analysis with code for DPKS. S.V. collected samples for the AKI data. A.L. provided input for the manuscript and supervision. J.M. provided supervision and contributed to the writing of the manuscript. L.M. oversaw and supervised the overall project and contributed to the writing of the manuscript. All authors approved the manuscript.

## Funding

## Competing interests

The authors declare no competing interests

## Inclusions and ethics

The AKI samples were processed in accordance with the Helsinki Declaration. The Ethics Committee of the Department of Surgery, Helsinki and Uusimaa Hospital District, approved the study protocol, and each participant or their proxy gave written informed consent. The Ethics Committee of the Department of Surgery, Helsinki and Uusimaa Hospital District, also approved the inclusion of participants for all centers involved as well as the use of deferred consent (Reference Number 18/13/03/02/2010). Patient demographics, medical history, severity scores, length of stay, physiologic data, and hospital mortality were collected from the Finnish Intensive Care Consortium prospective database (Tieto Ltd, Helsinki, Finland) with a study-specific case report form.
