## [Peer Review File · Communications Biology]

Reviewers' comments:

Reviewer #1 (Remarks to the Author):
(also attached)

Generalized peakgroup scoring boosts identification rates and accuracy in mass spectrometry based discovery proteomics

Summary

In the study "Generalized peakgroup scoring boosts identification rates and accuracy in mass spectrometry based discovery proteomics", Scott et al. propose an alternative generalizable validation method for DIA data. They build a denoising algorithm that can allow to filter out false positives and achieve more clean training data so that a subsequent percolator approach can boost identifications.

Review

Overall, I think the study by Scott et al. has some new and interesting ideas that could be of great value to the community, and I would, in principle, consider this a suitable study for Communications Biology. However, I found it difficult to follow the study, and after repeatedly going through the manuscript, I am still not entirely sure how some key mechanics of this study are working. I would recommend revising the manuscript in some paragraphs for clarity. Given the innovative nature of certain concepts, it would be unfortunate if they were not understood and appreciated.

Data used for Training

I find it difficult to understand what data the final model has seen. From Figure 1a) I would assume that the UMAP refers to the entire dataset ("of the unfiltered training labels"), so the 1.51M + 1.48M data points. There is an arrow to a confusion matrix, with no additional annotation, so nothing should change, but the confusion matrix has only ~ 750k data points. If this is the test split, please denote this in the figure. If I follow the arrow to the right, I see that only a subset of the data is split into training and held-out data but follows all the way to the denoised target labels. If I take a split of 0.2, this could add up to the 750k data points; however, the k-fold split would not make sense then. How exactly are the k-fold splits implemented, and how is the ensemble of classifiers constructed? See also the description of the denoising algorithm: "Each fold is scored by training an ensemble of [...] classifiers. [...] The ensemble of classifiers is then used to score the k-fold data." Is k-fold data referring to a split or the entirety of the data?

This could be addressed by having a flowchart about which data goes where and a more detailed depiction of the k-fold splits, ensemble classifier, and voting. The denoising algorithm consists of individual classifiers, and two additional classifiers based on output for the classifier are trained for evaluation. Training data can then either mean training data for the denoising algorithm or the evaluation classifier, so here, being explicit (e.g., calling this denoising training data and evaluation classifier training data) would help.

Method Benchmark Comparison

The method benchmark comparison provides interesting insight into the benefit of the new model when being in an entire workflow. However, at this point in the manuscript, I would have hoped to see some more technical benchmarking first, such as proposed below in Generalizability. I would suggest arranging the sections accordingly. Analyzing how the number of peptides changes within a log fold change is an interesting metric; however, a key benefit of a mixed-species experiment is that it nicely shows wrong identifications. Arguably, the peak of false positives for GPS seems the largest, and when

showing the precision of the method, the authors should also quantify the number of false positives (i.e., within the log₂ window). In general, the histograms a,c,e are hard to read. As there are y-labels, I would recommend removing having the same y-axis or using a log scale so that the yeast peak is more visible.

PIT analysis

The motivation of the PIT analysis is stated to allow more identifications at the same FDR, yet only relative changes in the respective rates are shown. Some insight into the actual number of additional identifications would be helpful.

Generalizability

The study aims to provide a generalized model. Currently, one insightful metric to demonstrate this is the UMAP plot of Figure 1a /c. This is a very qualitative metric, and I wonder if this could be more quantitative, e.g., by showing score distributions as in Fig 1d and e for different probability cutoffs? I would expect the false positive peak to disappear for increasing probability cutoffs.

The generalizability is somewhat implicitly shown by training on yeast data and later using this, e.g., for mouse or human data. However, for me to better follow the idea, technical benchmarking of this should already happen at an earlier stage, e.g., by showing the effect on the score distributions. Most sections introduce multiple new concepts at once, and one at a time would make it easier to follow. Figure 3 a shows a distribution of what is identified as False Target by the algorithm to showcase its accuracy, but where does the DScore come from? Ideally, I would want to see a systematic comparison of what happens to the target/decoy distributions for different species, as depicted in 1d/e.

AKI dataset

For the AKI dataset, the authors use MSFragger-DIA and this transition is not immediately clear, after the previous paragraphs were solely focusing on openSwath. This is also reflected on the GitHub, which states that: "Current support is for OpenSwath, but could be expanded to other tools quite easily.", so some additional context would be beneficial. On this note, other tools like DIA-NN or MaxDIA are not mentioned. While I think demonstrating the increment over the previous openSWATH is sufficient for publication, I think this will limit the potential userbase, and giving a perspective on the use with other tools would be appreciated.

ML & SHAP

The machine learning-aided differential expression is an interesting addition but ultimately distracting as it shifts the focus on the biological interpretation of the data and not the benefit gained by the GPS method. I would therefore suggest removing this part or alternatively showcasing how applying the GPS method would change the SHAP values, as in Figure 4, with the differential expression.

Misc

- The authors compare openSWATH, other tools like DIA-NN or MaxDIA are not mentioned. While I think demonstrating the increment over the previous openSWATH is sufficient for publication, I think this will limit the potential user base.
- The discussion mentions the pitfalls of low-n methods and that the approach is aimed to minimize this. I fail to see where the benefits of low-n data are shown in the manuscript, and I would suggest including this or rewriting this paragraph.

- Using a comma-separator for thousands (1278834 -> 1,278,834) would increase readability
- The introduction mentions several landmark papers for DL spectra prediction. However, the first to demonstrate this was Zhou et al., Anal Chem., 2017 (pDeep), so I would suggest including this study as well.
- The introduction suggests that mProphet is "the algorithm of choice" in the validation of DIA mass spectrometry data, which seems like a general statement about DIA analysis, yet only OpenSwath is mentioned. Other DIA-capable tools, such as DIA-NN or MaxDIA are not referenced. I suggest either addressing how these other tools validate DIA data or rephrasing this sentence to make it clear that this is for the OpenSwath case.
- Arguably, the score distributions are more important than the absolute values as the subsequent tools will determine a cutoff based on them. I would therefore suggest to not having the same x-limits for d) and e) in Figure 1 but rather adjusting them so that the distributions can be compared.
- MSFragger-DIA is referenced by two references, but none of the references points to the 2022 bioRxiv manuscript where it was introduced.
- The bibliography seems to be erroneous, e.g., it sometimes contains the month in lowercase.
- In Fig 2b, GPS is green in the three-color scheme, and in 4 It is dark blue. A consistent scheme would help.

Reviewer #2 (Remarks to the Author):

The authors proposed a generalizable validation method (GPS) that could confidently control FDR while increasing the number and accuracy of identifications in DIA MS. They focused on an important issue for DIA data analysis. I like their idea. But, the description is not very clear. And I have some concerns about their methods and results.

Major comments:

1. How did the authors eliminate the overfitting risk in their model?
2. How did the authors handle the imbalanced datasets?
3. Why did the authors say that there is no need to optimize the parameters of GPS?
4. The results did not show a substantive improvement over existing tools. For example, the results in Figure 2f seem to show that the precision of GPS and Pyprophet are the best. And as shown in Figure 3, the performance of GPS is comparable with Pyprophet.
5. The results shown in Figure 4d-f seems odd to me. GPS did not identify more proteins, but it identifies the most DE proteins. I think this could not show the advantage of GPS.

Minor comments:

1. The line colors for GPS, Pyprophet, Percolator in Figure 2 and Figure 3 is not easy to distinguish.

Reviewer #3 (Remarks to the Author):

In "Generalized peakgroup scoring boosts identification rates and accuracy in mass spectrometry based discovery proteomics" the authors present their Generalizable Peakgroup Scoring (GPS) that is used to score peak groups in data-independent acquisition (DIA) proteomics experiments. The manuscript presents an interesting rescoring method for DIA data that is akin to PyProphet and Percolator. Although I have questions about the methodology and analysis, I would recommend this manuscript for publication after major revision.

Major Issues

-
1. The authors should define "peakgroups" precisely, as it is foundational to the manuscript. The first mention of the term in is the opening paragraph, "...resulting in low numbers of validated peakgroups and imprecise FDR control..." in a context that I normally associate with precursors or peptides.
 2. Data-dependent acquisition (DDA) experiments are regularly analyzed by searching against whole proteomes. Are the same sensitivity issues seen in the DDA setting? If not, why are they particular to DIA data?
 3. PyProphet and Percolator begin by assigning confident targets the positive label as assessed by thresholding at a particular FDR using the single best feature in the input data. How do the "true targets" selected by the proposed denoising procedure compare with these sets? One could, for example, plot the size of the intersection of the denoised positive labels with the positive labels that would be assigned at various FDR thresholds.
 4. The UMAP projections in Figure 1 are interesting. Have the authors compared the proposed denoising method to a 2-group clustering, say by K-means, in the original feature space? The structure in the UMAP suggests that this may work similarly.
 5. What features are used to describe each peakgroup?
 6. The manuscript indicates that a classifier is trained using the denoised labels to rescore peakgroups; however, it is unclear to me what form this classifier takes. If this classifier is a non-linear model, such as a random forest or gradient boosted machine, the increased power from GPS may be attributable to this quality alone. Such an instance would warrant an additional experiment wherein GPS is restricted to a linear model, such as logistic regression or a support vector machine with a linear kernel, because PyProphet and Percolator are both restricted to linear solutions.
 7. How is protein-level FDR estimated?
 8. I like the use of expected relative abundances to evaluate detection performance. Was the log₂ fold change analysis conducted at the peakgroup, peptide, or protein level?
 9. I found the PIT analysis section to be confusing. The way I read the section is that the results from the denoising procedure were used as ground truth for the remainder of the analysis, which would be problematic due to circular logic. However, the experiment appears to be setup as a standard entrapment experiment. Can the authors clarify this section? Such an analysis should be independent of the proposed methods in the manuscript. In most cases the entrapment FDR is calculated as the number of accepted entrapment sequences divided by the number of accepted target sequences, adjusted for the total number of entrapment sequences in comparison to the total number of target sequences.
 10. How are decoy peakgroups generated?
 11. I found the use of SHAP values from a classifier in place of a statistical differential expression test to be acceptable, but I found the reasoning behind this choice to be lacking. Fundamentally, the two methods yield different types of results: a statistical differential expression test assesses what analytes can be reproducibly and consistently measured as different between two conditions. In contrast, the SHAP values from a classifier indicate which features were important for the classification task - however, the features need not be consistent shifts in one direction, nor reproducible. For example,

high variance of a measured analyte may be indicative of a condition, such that any extreme value is predictive of the condition. Such an analyte may have a high importance by SHAP, but poor statistical significance. Additionally, although p-values are normally arbitrarily thresholded, they are continuous values and can be treated as such. In the ML literature, feature attribution is also moving toward statistical measures, such as with the introduction of "knockoffs" to calculate false-discovery rates among features (https://candes.su.domains/publications/downloads/FDR_regression.pdf).

Minor Issues

-
1. The opening sentence of the introduction states, "One disadvantage of data independent acquisition (DIA) proteomics is that a spectral library based on previously identified peptides in a sample is required to interpret the complex signal and quantify peptides." This is not strictly true, as there exist several library free approaches; for example, those that generate pseudo-spectra from correlated features, such as with DIAUmpire. Indeed, MSFragger-DIA is mentioned later in the manuscript.
 2. Should "peakgroup" be a single word or two ("peak group")? My inclination is the latter.
 3. When describing the class imbalance problem with Percolator, the manuscript states, "... some true targets are identified, they would represent a small fraction compared to the negative decoys in the data, creating a overwhelming class imbalance, which can destabilize the training of machine learning algorithms if not dealt with in an appropriate manner." This statement is indeed true. Percolator, specifically, attempts to address this by selecting the cost hyperparameter for each class separately over a grid search. This essentially functions as a class weight parameter, which can downweight importance of the negative, decoy class relative to the positive, target class. Note that the nested cross-validation procedure is described by Granholm et al. (<https://bmcbioinformatics.biomedcentral.com/articles/10.1186/1471-2105-13-S16-S3>)

Referee expertise:

Referee #1: machine learning, proteomics

Referee #2: bioinformatics, proteomics

Referee #3: machine learning, proteomics

Reviewers' comments:

Reviewer #1 (Remarks to the Author):
(also attached)

Generalized peakgroup scoring boosts identification rates and accuracy in mass spectrometry based discovery proteomics

Summary

In the study “Generalized peakgroup scoring boosts identification rates and accuracy in mass spectrometry based discovery proteomics”, Scott et al. propose an alternative generalizable validation method for DIA data. They build a denoising algorithm that can allow to filter out false positives and achieve more clean training data so that a subsequent percolator approach can boost identifications.

Review

Overall, I think the study by Scott et al. has some new and interesting ideas that could be of great value to the community, and I would, in principle, consider this a suitable study for Communications Biology. However, I found it difficult to follow the study, and after repeatedly going through the manuscript, I am still not entirely sure how some key mechanics of this study are working. I would recommend revising the manuscript in some paragraphs for clarity. Given the innovative nature of certain concepts, it would be unfortunate if they were not understood and appreciated.

Data used for Training

I find it difficult to understand what data the final model has seen. From Figure 1a) I would assume that the UMAP refers to the entire dataset (“of the unfiltered training labels”), so the 1.51M + 1.48M data points. There is an arrow to a confusion matrix, with no additional annotation, so nothing should change, but the confusion matrix has only ~ 750k data points. If this is the test split, please denote this in the figure. If I follow the arrow to the right, I see that only a subset of the data is split into training and held-out data but follows all the way to the denoised target labels. If I take a split of 0.2, this could add up to the 750k data points; however, the k-fold split would not make sense then. How exactly are the k-fold splits implemented, and how is the ensemble of classifiers constructed? See also the description of the denoising algorithm: “Each fold is scored by training an ensemble of [...] classifiers. [...] The ensemble of classifiers is then used to score the k-fold data.” Is k-fold

data referring to a split or the entirety of the data?

This could be addressed by having a flowchart about which data goes where and a more detailed depiction of the k-fold splits, ensemble classifier, and voting. The denoising algorithm consists of individual classifiers, and two additional classifiers based on output for the classifier are trained for evaluation. Training data can then either mean training data for the denoising algorithm or the evaluation classifier, so here, being explicit (e.g., calling this denoising training data and evaluation classifier training data) would help.

The introductory figure has been updated to better represent the flow of the data in the training of the GPS static models and to demonstrate the layout of the study (see new Figure 1 in the manuscript) and to remove clutter and confusion. We also have clarified in the text to help make sense of what is going on with the training data and the held out test data. The data is split first into a training set and a test set, where then the training set is first used to train a classifier using the noisy labeled data. That same training data is then run through the denoising algorithm to train a second classifier on the denoised data and these are compared. The function of the denoising algorithm is to filter out false positive labels in a way that still leads to models that can easily generalize to new data. The denoising algorithm trains on the training data itself using k-folds, so that each ensemble of classifiers vote on data that they have not seen during training. In this sense, the algorithm is self denoising, in that it does not need external training data. This has been described in depth in the Methods section, but essentially the denoising algorithm takes the training split from above (2,988,116 peak groups), and splits into 10-folds. For each fold, the remaining data is used to train an ensemble of classifiers using bagging that then vote on the held out fold of data so that they are not voting on data that is used to train the ensemble. If all of the classifiers in the ensemble vote (ie. Classify) that the peak group is a true target, then that target is kept in the filtered training set. This procedure results in 2,754,877 peakgroups. There are now 2 training datasets, one that is unfiltered and one that is filtered, and these are used to train both an SVM and XGBoost model, resulting in 4 GPS models that are then used for verification.

The text in the results has been updated in the following manner in an attempt to help clarify what is going on in the Results section under the “GPS effectively generalizes to new data.” header:

“The first step in evaluating GPS is to establish a method for training models that will generalize effectively to new data. To that end, 129 samples of the yeast data were randomly split into 102 training samples and 26 test samples set aside to be used for validation and to assure the trained models are not over-fitted. The combined data consists of 3,751,367 peak groups while the 102 training samples consist of 2,988,116 peak groups (1,479,571 decoys and 1,508,545 targets), and we will refer to this data as the unfiltered training data. The test samples consist of 763,251 peak groups, and we will refer to them as the yeast test data. The unfiltered training data was filtered using a novel denoising algorithm to remove noisy false target labels that destabilize model training. This denoising algorithm takes a sample as input and initially splits the precursors in the sample into 10-folds. So that each held-out fold can be scored using classifiers trained on separate data, the

remaining data is used to train an ensemble of 10 logistic regression classifiers with bagging [30]. Each classifier in the ensemble will vote on the held-out data and only precursors where every classifier in the ensemble votes it as a true target are kept in the training data. This filtering resulted in a training set of 2,754,877 peak groups (1,479,571 decoys and 1,275,306 targets) we will refer to as the filtered training data. Linear SVM and non-linear XGBoost models were trained using each training set to create 4 models (XGB Filter, SVM Filter, XGB No Filter, SVM No Filter). Additionally, we trained a PyProphet XGB and a PyProphet LDA model on the unfiltered training data as a comparison to the GPS training method. These 6 models were applied to evaluate the effects of training set filtering and model type on model precision, generalization to new data, and the ability to maximize identifications. These models were evaluated on 3 distinct data sets to confirm they generalize and to ensure that overfitting of the models is not occurring.”

Method Benchmark Comparison

The method benchmark comparison provides interesting insight into the benefit of the new model when being in an entire workflow. However, at this point in the manuscript, I would have hoped to see some more technical benchmarking first, such as proposed below in Generalizability. I would suggest arranging the sections accordingly. Analyzing how the number of peptides changes within a log fold change is an interesting metric; however, a key benefit of a mixed-species experiment is that it nicely shows wrong identifications. Arguably, the peak of false positives for GPS seems the largest, and when showing the precision of the method, the authors should also quantify the number of false positives (i.e., within the log₂ window). In general, the histograms a,c,e are hard to read. As there are y-labels, I would recommend removing having the same y-axis or using a log scale so that the yeast peak is more visible.

The initial figure of analysis is now focused on how the GPS models can generalize effectively to 3 different datasets (Figure 2). This includes the number of ID's at increasing FDR thresholds, the measured precision for each of the 4 classifiers on each unrelated dataset. Their score distributions and confusion matrices are available as Suppl. Figure 1 and 2.

The raw number of false positives identified is important, but arguably the FDR, or the ratio of false positives to true positives at certain cutoffs is more important than the raw counts. Figure 4F depicts directly the FDR (ie. The number of false peaks that fall within increasing thresholds around the expected ratios). This shows that all tools in the comparison have comparable FDRs, but GPS identifies many more true positives while maintaining the same FDR as PyProphet. Additionally, the histograms of Figure 4A-B have also been zoomed in to show that the false positive peak for GPS is not the largest. The histograms have also been replaced with MA plots to better show the distribution of ID's over abundance with a zoom into the densities of the distributions with the histograms to increase interpretability.

PIT analysis

The motivation of the PIT analysis is stated to allow more identifications at the same FDR, yet only relative changes in the respective rates are shown. Some insight into the actual number of additional identifications would be helpful.

The PIT analysis has been altered to focus on the entrapment FDR to show how many identifications are passed in at increasing FDR thresholds (Figure 3C) and how the Yeast FDR (which is the rate of Yeast IDs) increases with increasing “real” FDR thresholds (Figure 3D). Upon reanalysis, it was found that PIT estimation is detrimental to quantification accuracy, so this comment was useful in identifying this. Instead of using PIT to boost identification numbers, we can instead utilize the predictive power of GPS to accurately filter data and control the FDR.

Generalizability

The study aims to provide a generalized model. Currently, one insightful metric to demonstrate this is the UMAP plot of Figure 1a /c. This is a very qualitative metric, and I wonder if this could be more quantitative, e.g., by showing score distributions as in Fig 1d and e for different probability cutoffs? I would expect the false positive peak to disappear for increasing probability cutoffs.

The UMAP was used to show that the training labels were noisy and not that the models would generalize to new data as we agree that this is a qualitative metric. In response, these UMAPs proved to be confusing in communicating this idea, so they were removed and a graphical abstract figure to show the overall GPS framework was included (Figure 1A). To show how models trained on the different datasets generalize to new data, we have expanded Figure 2 to focus on the evaluation of the 4 GPS models, and 2 PyProphet models, on 3 distinctly different test data sets. The generalization performance is now directly measured by the number of ID's that pass at increasing FDR thresholds, as well as the precision of the GPS models on the 3 different test data sets. Additionally, the score distributions for all models as been added as Suppl. Figure 2 and the confusion matrices for all models has been added as Suppl. Figure 1. These metrics were added to provide a quantitative outlook on how well these models generalize to new unrelated data.

The generalizability is somewhat implicitly shown by training on yeast data and later using this, e.g., for mouse or human data. However, for me to better follow the idea, technical benchmarking of this should already happen at an earlier stage, e.g., by showing the effect on the score distributions. Most sections introduce multiple new concepts at once, and one at a time would make it easier to follow. Figure 3 a shows a distribution of what is identified as False Target by the algorithm to showcase its accuracy, but where does the DScore come from? Ideally, I would want to see a systematic comparison of what happens to the target/decoy distributions for different species, as depicted in 1d/e.

Target Decoy distributions have been added for 3 different species datasets, and included in the analysis for model generalization and evaluation as seen in Figure 2. These Target Decoy distributions for the Yeast, Mouse, and Human datasets have been added as Suppl. Figure 2. The analysis of GPS on the 3 test data sets was moved to the first part of the results to evaluate how GPS can generalize to new data (Figure 2).

Figure 3A-B shows scores on the x-axis as the output from GPS (ie. The combined score, decision score, etc.), as well as the False target distribution as Yeast identifications in the library. This has been clarified in the text and figure description as well.

“The effects of peak group predictions are evident in their removal of yeast identifications from consideration and the decrease in the false target portion of the bimodal target distribution of GPS output scores (**Figure4A-B**)”

AKI dataset

For the AKI dataset, the authors use MSFragger-DIA and this transition is not immediately clear, after the previous paragraphs were solely focusing on openSwath. This is also reflected on the GitHub, which states that: “Current support is for OpenSwath, but could be expanded to other tools quite easily.”, so some additional context would be beneficial. On this note, other tools like DIA-NN or MaxDIA are not mentioned. While I think demonstrating the increment over the previous openSWATH is sufficient for publication, I think this will limit the potential userbase, and giving a perspective on the use with other tools would be appreciated.

The use of MSFragger-DIA, see

https://fragpipe.nesvilab.org/docs/tutorial_fragpipe_workflows.html#msfragger-dia-wide-window-speclib), was to augment the human tissue spectral library with identifications from the direct DIA search as well as correct the retention time. The resulting library was analyzed using OpenSwath and GPS in the same way as the rest of the analysis was performed. This was updated in the Results section of the text under the Application header to avoid any confusion as the idea was to optimize the spectral library.

“We interrogated this data using an optimized human tissue spectral library consisting of the Pan Human Library [5] and appended spectra from direct DIA identifications using MSFragger (v3.5) [32] to correct the retention time and augment the library with more identified precursors (10952 proteins overall). This analysis puts into context the benefits that GPS provides when querying a large search space and the benefit of using extensive curated repository spectral libraries in discovery DIA.”

We have added text to the introduction to mention tools such as DIA-NN and MaxDIA, however these are closed source tools that have no options to output the subscores that are calculated to assign peak quality, train their machine learning models, and control the false discovery rate. However, we do believe that if these tools applied a strategy of peak group prediction and static model scoring as with GPS, then they could also see many of the benefits that we have through optimizing OpenSwath. Theoretically, GPS can be adapted to

work easily with any open source tool that exposes the subscores that are calculated and allow for static model training, peak group prediction, and rescoring. The main idea behind the analysis in this manuscript was to showcase the theory behind GPS and demonstrate it's benefits in a direct comparison (ie. to OpenSwath with PyProphet).

ML & SHAP

The machine learning-aided differential expression is an interesting addition but ultimately distracting as it shifts the focus on the biological interpretation of the data and not the benefit gained by the GPS method. I would therefore suggest removing this part or alternatively showcasing how applying the GPS method would change the SHAP values, as in Figure 4, with the differential expression.

The point of the AKI analysis was to showcase how GPS could be used in an end-to-end discovery DIA analysis to identify a group of potential proteins that could be used to answer a real biological question. Our idea was to first showcase the technical aspects of GPS on benchmarking datasets, and then show how GPS can be applied in a real world setting. Taking this comment into consideration, we have combined Figure 4 and 5, and downplayed the importance of the SHAP analysis and biomarker discovery in section 2.5 of the Results, "The application of GPS to identify potential protein biomarkers for sepsis induced AKI."

"Using the increased analytical depth of GPS, we applied recursive feature elimination (RFE) with cross validation via explainable artificial intelligence (SHAP[34]) and an XGBoost classifier to identify a panel of highly accurate predictive proteins in differentiating between AKI subphenotypes. SHAP-RFE analysis identified a group of 18 proteins as the most effective discriminatory panel (Figure5D)."

Figure 5A-C depict how GPS can outperform PyProphet when analyzing undepleted plasma samples, and then Figure 5D-F depict how these improvements can manifest in the discovery of potentially relevant panels of biomarkers.

Misc

- The authors compare openSWATH, other tools like DIA-NN or MaxDIA are not mentioned. While I think demonstrating the increment over the previous openSWATH is sufficient for publication, I think this will limit the potential user base.

As mentioned above, this is unfortunately impossible as MaxDIA and DIA-NN are closed source tools that do not expose the scores that are calculated for each precursor, so therefor it is impossible to extend those tools to use GPS. However, as we have seen the benefits of GPS with OpenSwath, we hypothesize that MaxDIA and DIA-NN would both benefit from implementing a generalizable scoring algorithm such as GPS that uses static models trained on a curated dataset. This has been added to the Discussion:

Ideally, we would have liked to use the subscores calculated by DIA-NN to train a GPS classifier and then apply the GPS framework within the DIA-NN tool-chain, but since DIA-NN

is closed source and does not expose or record the subscores calculated to estimate precursor quality, this direct comparison and integration was not possible. Given these improvements, we were still able to bring GPS close to the performance of DIA-NN on the spike-in data in terms of overall identifications, and even surpass DIA-NN when it comes to decreasing the number of missing values and higher quantitative accuracy closer to the expected ratios of the spike-in data (**Supplementary Figure S3**). We believe that if DIA-NN implemented a strategy where a static model was used to predict and score precursors, that some of the same benefits we see in the OpenSwath pipeline could be realized to further improve identification and quantification with DIA-NN.”

Although it is not a fair comparison due to the superior signal processing and different subscores calculated by DIA-NN, we have compared GPS and DIA-NN performances by identification counts at 1% FDR on end-result data in Supplementary Figure 3. This analysis demonstrates how GPS can achieve close to state-of-the-art performance even with the improved signal processing algorithms used in DIA-NN over OpenSwath. Notably, GPS is able to decrease the number of missing values compared to DIA-NN and identify more precursors closer to the expected ratios at low fold change thresholds.

- The discussion mentions the pitfalls of low-n methods and that the approach is aimed to minimize this. I fail to see where the benefits of low-n data are shown in the manuscript, and I would suggest including this or rewriting this paragraph.

The text we included in the discussion was erroneous and has been removed. We were referring to this study (<https://doi.org/10.1021/acs.jproteome.9b00780>) by Fondrie et al that describes how static models can help boost identification in experiments with low sample numbers. This is low-n in that there are low sample numbers involved in the study, but this was just mentioned as scenario that generalizable models such as GPS could help in.

- Using a comma-separator for thousands (1278834 -> 1,278,834) would increase readability

These have been updated.

- The introduction mentions several landmark papers for DL spectra prediction. However, the first to demonstrate this was Zhou et al., Anal Chem., 2017 (pDeep), so I would suggest including this study as well.

This reference has been added.

- The introduction suggests that mProphet is “the algorithm of choice” in the validation of DIA mass spectrometry data, which seems like a general statement about DIA analysis, yet only OpenSwath is mentioned. Other DIA-capable tools, such as DIA-NN or MaxDIA are not referenced. I suggest either addressing how these other tools validate DIA data or rephrasing this sentence to make it clear that this is for the OpenSwath case.

We had referenced that this is an expansion to OpenSwath in the discussion, but this is now emphasized in the introduction to make sure. It is also mentioned that these other tools all use iterations of the mProphet algorithm. Spectronaut, DIA-NN (selects positive training

examples and then uses a small ensemble of neural networks to score the data), EncyclopeDIA (which uses Percolator directly), MaxDIA, all train sample specific/experiment specific classifiers in a similar manner to mProphet. Citations to DIA-NN and MaxDIA have been added to the introduction and the text has been updated accordingly.

“Apart from the OpenSWATH toolchain, more modern approaches such as DIA-NN [20] utilize a modified version of the mProphet algorithm, where positive targets are selected based on an initial FDR cutoff and then an ensemble of neural networks are trained to classify target and decoy peak groups. Additionally, EncyclopeDIA[21] directly uses Percolator[17] to validate peak groups, while MaxDIA[22] trains an XGBoost classifier for each experiment to control the FDR. All of these common DIA analysis tools utilize some aspects of the mProphet algorithm and rely on training new classifiers for each subsequent analysis, which can be a computationally heavy and non-trivial task”

- Arguably, the score distributions are more important than the absolute values as the subsequent tools will determine a cutoff based on them. I would therefore suggest to not having the same x-limits for d) and e) in Figure 1 but rather adjusting them so that the distributions can be compared.

This has been done and all score distributions can be visualized on their own x-axis without constraints in Suppl. Figure 2.

- MSFragger-DIA is referenced by two references, but none of the references points to the 2022 bioRxiv manuscript where it was introduced.

This was after the initial writing of the manuscript. The reference has been updated where applicable.

- The bibliography seems to be erroneous, e.g., it sometimes contains the month in lowercase.

It is automatically formatted by LaTeX from BibTex references. All months were corrected.

- In Fig 2b, GPS is green in the three-color scheme, and in 4 It is dark blue. A consistent scheme would help.

A consistent color scheme has been updated for the tools, methods, and data compared throughout the manuscript.

Reviewer #2 (Remarks to the Author):

The authors proposed a generalizable validation method (GPS) that could confidently control FDR while increasing the number and accuracy of identifications in DIA MS. They focused on an important issue for DIA data analysis. I like their idea. But, the description is not very clear. And I have some concerns about their methods and results.

Major comments:

1. How did the authors eliminate the overfitting risk in their model?

We eliminated the risk of overfitting first by splitting the training data into a train and test set so GPS could be evaluated on data it had not seen during training. Additionally, we tested our model on 2 other distinctly different test data sets that the models had not seeing during training. The GPS models performed exceedingly well on all models when comparing the metric (precision score) that was most important in ensuring that only true targets were predicted. This is visualized in Figure 2 and Suppl. Figure 1.

2. How did the authors handle the imbalanced datasets?

The small imbalance in the dataset was handled by calculating this class imbalance and passing these ratios to the GPS training algorithms so that each instance in the dataset was weighted correctly. We have clarified this in the Methods section under “Model training”, and added the following to the discussion:

“In the case of GPS, instead of choosing to implement imbalanced learning methods with sample specific classifiers we trained models on curated data that generalize to unrelated samples, so that it does not matter if there is a class imbalance or noisy labels in the data being predicted and score. To ensure that the no class imbalance issues effect training, we calculate the class ratios and pass them to our training algorithms so that weights are adjusted accordingly and the imbalance is taken into account during training.”

3. Why did the authors say that there is no need to optimize the parameters of GPS?

Since GPS consists of an already trained model, to apply GPS to a new dataset, you just need to use this existing model to predict peak groups and score the data. Other software may need extensive configuration and hyper-parameter tuning as new models are trained for every experiment, which is especially true if the data set is non-standard. Text has been added to the Discussion to address this in a clear manner.

“Additionally, there is no need to optimize hyperparameters to squeeze the best performance out of GPS, as the generalized model will predict and score new precursors accurately in a stable manner no matter the conditions of the data. PyProphet can be optimized to train sample specific classifiers in different conditions, but a meticulous process to find optimal hyperparameters is required. This non-trivial and time consuming parameter optimization and model training can be avoided completely by applying generalized models, such as GPS, directly to new data as they have already been trained and evaluated.”

4. The results did not show a substantive improvement over existing tools. For example, the results in Figure 2f seem to show that the precision of GPS and Pyprophet are the best. And as shown in Figure 3, the performance of GPS is comparable with Pyprophet.

Figure 2F only showed the precision of GPS and PyProphet, but was important in the context that GPS identifies far more precursors, peptides, and proteins than PyProphet, overall and

as ratio validated identifications but maintains the same precision. This was mentioned in the text that the precision of GPS remains very high, and comparable to PyProphet, while still identifying and quantifying more proteins than PyProphet. As per other comments from reviewers, Figure 2 has been moved to Figure 4, and we can see that GPS has more ID's closer to their expected ratios (Figure 4C), identifies 18.97% more precursors, 17.96% more peptides, and 5.28% more proteins than PyProphet, and has a comparable FDR throughout all measured cutoffs while identifying more precursors (Figure 4E-F). Overall, Figure 4 shows that in a direct comparison to PyProphet, GPS identifies thousands of more precursors while maintaining a comparable FDR.

5. The results shown in Figure 4d-f seems odd to me. GPS did not identify more proteins, but it identifies the most DE proteins. I think this could not show the advantage of GPS.

Since we are mostly interested in how many proteins we could compare between 2 groups, the tool that identifies the most comparable proteins is the best performing. This analysis has been updated in Figure 5C to show that GPS identifies more proteins overall, more proteins that are quantified in at least 10 replicates per group, and more proteins that are considered statistically significant (p -value < 0.1) than PyProphet. We removed the comparison to Percolator as we did not believe it a fair and direct comparison, and it resulted in a matrix with the most missing values, and a very low number of comparable proteins.

Minor comments:

1. The line colors for GPS, Pyprophet, Percolator in Figure 2 and Figure 3 is not easy to distinguish.

The line colors for all figures have been updated to help with readability.

Reviewer #3 (Remarks to the Author):

In "Generalized peakgroup scoring boosts identification rates and accuracy in mass spectrometry based discovery proteomics" the authors present their Generalizable Peakgroup Scoring (GPS) that is used to score peak groups in data-independent acquisition (DIA) proteomics experiments. The manuscript presents an interesting rescoring method for DIA data that is akin to PyProphet and Percolator. Although I have questions about the methodology and analysis, I would recommend this manuscript for publication after major revision.

Major Issues

1. The authors should define "peakgroups" precisely, as it is foundational to the manuscript. The first mention of the term in is the opening paragraph, "...resulting in low numbers of validated peakgroups and imprecise FDR control..." in a context that I normally associate with precursors or peptides.

In order to avoid confusion, we have replaced “peakgroups” with precursors throughout the manuscript. A peak group is indeed a precursor, so switching the nomenclature does not change the concepts in the manuscript.

2. Data-dependent acquisition (DDA) experiments are regularly analyzed by searching against whole proteomes. Are the same sensitivity issues seen in the DDA setting? If not, why are they particular to DIA data?

Yes, but less so. The number of potential IDs grows for DDA with larger protein sequence libraries, but since you have a finite number of spectra, and are measuring the scores using those tens of thousands of spectra, the issue is not as bad. Typical DIA analysis will extract chromatograms for every entry in the spectral library, so if you have samples that maybe do not match the library, such as plasma samples and large tissue libraries, many of the chromatograms stated as true will not actually be in the sample, so the assumptions are different. In DDA it is assumed that a match from the spectra to the library could be true, but in DIA with plasma samples, you would assume most extracted peakgroups are false positives. The same issues are seen in DDA in proteogenomics (<https://academic.oup.com/bib/article/23/5/bbac163/6582880?login=true>) where the search space is exorbitantly large, and in that case the issue is similar to DIA. We have added text to the Introduction to address this:

“A large search space is also prevalent in proteogenomic experiments (ie. searching for single amino acid variants) or the sequencing of antibodies from pull-down experiments, where the variation in potential protein sequences causes the search space to sky-rocket [23].”

3. PyProphet and Percolator begin by assigning confident targets the positive label as assessed by thresholding at a particular FDR using the single best feature in the input data. How do the “true targets” selected by the proposed denoising procedure compare with these sets? One could, for example, plot the size of the intersection of the denoised positive labels with the positive labels that would be assigned at various FDR thresholds.

This is an interesting idea for a comparison. Instead of directly comparing the overlap of these precursors, we instead directly compared GPS trained on denoised data directly to models trained with PyProphet on the same data and analyzed how they generalized to new data (Figure 2). We believe that this comparison would be interesting for analysis in a separate study, but felt it did not directly contribute to the overall story we wanted to convey in the manuscript.

4. The UMAP projections in Figure 1 are interesting. Have the authors compared the proposed denoising method to a 2-group clustering, say by K-means, in the original feature space? The structure in the UMAP suggests that this may work similarly.

The UMAP projection is not the entire dataset (only 50,000 data points), and the performance is significantly worse when this is increased above 50,000. It’s possible that the entire dataset may look different when processed and projected down into 2 dimensions also. Theoretically this would be extremely interesting to try, but the practicality is not so

evident. The benefit of the denoising algorithm is that it is memory and CPU efficient, and can be efficiently parallelized making it more practical. It is

In response to additional comments from other reviewers, the UMAP visualizations were removed from the manuscript. The UMAP was used to show that the training labels were noisy, but these UMAPs proved to be confusing in communicating this idea, so they were removed and a graphical abstract figure to show the overall GPS framework was included (Figure 1A).

5. What features are used to describe each peakgroup?

The features used to describe each peak group in this study are the MS2 features calculated by OpenSwath. They are as follows: var_bseries_score, var_dotprod_score, var_intensity_score, var_isotope_correlation_score, var_isotope_overlap_score, var_library_corr, var_library_dotprod, var_library_manhattan, var_library_rmsd, var_library_rootmeansquare, var_library_sangle, var_log_sn_score, var_manhattan_score, var_massdev_score, var_massdev_score_weighted, var_mi_score, var_mi_weighted_score, var_norm_rt_score, var_xcorr_coelution, var_xcorr_coelution_weighted, var_xcorr_shape_weighted, var_yseries_score

This has been updated in the text in the Methods section:

“The model details are available at <https://github.com/InfectionMedicineProteomics/gps>. Input features for the training of the models are based on MS2 level subscores as calculated by OpenSwath and are as follows: var_bseries_score, var_dotprod_score, var_intensity_score, var_isotope_correlation_score, var_isotope_overlap_score, var_library_corr, var_library_dotprod, var_library_manhattan, var_library_rmsd, var_library_rootmeansquare, var_library_sangle, var_log_sn_score, var_manhattan_score, var_massdev_score, var_massdev_score_weighted, var_mi_score, var_mi_weighted_score, var_norm_rt_score, var_xcorr_coelution, var_xcorr_coelution_weighted, var_xcorr_shape_weighted, var_yseries_score.”

6. The manuscript indicates that a classifier is trained using the denoised labels to rescore peakgroups; however, it is unclear to me what form this classifier takes. If this classifier is a non-linear model, such as a random forest or gradient boosted machine, the increased power from GPS may be attributable to this quality alone. Such an instance would warrant an additional experiment wherein GPS is restricted to a linear model, such as logistic regression or a support vector machine with a linear kernel, because PyProphet and Percolator are both restricted to linear solutions.

This was a great comment and warranted a major revision of the analysis, which has been rerun with this in mind. We have trained 4 GPS classifiers using SVM and XGBoost models on both the filtered and unfiltered training data, and directly compared those to PyProphet LDA and PyProphet XGBoost classifiers (Figure 2). After that initial analysis, all other comparisons in the manuscript are done between GPS XGB Filter and the PyProphet XGB model. PyProphet XGB models performed much better than the LDA models used in the initial analysis, but the generalizing predictive power of GPS is able to outperform

PyProphet in the measured comparisons (Figure 2A-C). This suggestion was crucial to the reorganization of the manuscript and led to an overall more coherent story as we can directly compare the 2 methods. Additionally, we show that the predictive power of GPS can be used to further boost performance compared to PyProphet XGB as seen in the comparisons in Figure 3-5. After the initial comparison, we have clarified in the Generalization subsection of the Results that the GPS XGB Filter is used throughout the rest of the manuscript and referred to as GPS:

“Due to the superior number of identifications passing 1% on the test data and the highest precision score among classifiers, the GPS XGB Filter classifier will be used for the remainder of the study and referred to as GPS.”

7. How is protein-level FDR estimated?

Protein and peptide level FDR is estimated using the picked-ID approach at the global level based on the PyProphet implementation in Rosenberger et al (<https://www.nature.com/articles/nmeth.4398>). A graph connecting precursors to peptides and proteins is used to select the highest scoring precursor per peptide across all runs and the highest scoring precursor per protein across all runs. Q-values are then calculated using the resulting score distributions. This has been clarified in the Methods section of the manuscript.

“Global FDR control is implemented in a similar manner to PyProphet[18], where all scored samples in an experiment are aggregated and the highest scoring precursor is selected to represent either the peptide or the protein at the desired level. Once the highest scoring precursors are selected, q-values are estimated using the method described above. The resulting scoring models are exported as serialized Python objects that can then easily be used from the command line by GPS to export an annotated quantitative matrix.”

8. I like the use of expected relative abundances to evaluate detection performance. Was the log2 fold change analysis conducted at the peakgroup, peptide, or protein level?

This was done at the precursor level as we wanted to see the effects of the different validation methods on the quantified signal with obfuscating the data by rolling up to protein quantities. This has been clarified in the text in the Quantification subsection of the Results.

9. I found the PIT analysis section to be confusing. The way I read the section is that the results from the denoising procedure were used as ground truth for the remainder of the analysis, which would be problematic due to circular logic. However, the experiment appears to be setup as a standard entrapment experiment. Can the authors clarify this section? Such an analysis should be independent of the proposed methods in the manuscript. In most cases the entrapment FDR is calculated as the number of accepted entrapment sequences divided by the number of accepted target sequences, adjusted for the total number of entrapment sequences in comparison to the total number of target sequences.

Due to this suggestion we have substantially reworked this section. This was, in fact, a standard entrapment FDR analysis, which is clarified in the Identification subsection of the Results and updated in Figure 3. We wanted to show how the precision of GPS can be used to predict true target precursors, increase the number of correct IDs, and eliminate false yeast identifications from contention. The Yeast FDR was calculated the same way as the “normal” FDR, which is the method you suggested.

10. How are decoy peakgroups generated?

Decoy peakgroups are generated using OpenSwathDecoyGenerator from OpenMS (v3.0) with default parameters using the shuffle method. This has been added to the Methods section “Spectral library creation”.

11. I found the use of SHAP values from a classifier in place of a statistical differential expression test to be acceptable, but I found the reasoning behind this choice to be lacking. Fundamentally, the two methods yield different types of results: a statistical differential expression test assess what analytes can be reproducibly and consistently measured as different between two conditions. In contrast, the SHAP values from a classifier indicate which features were important for the classification task - however, the features need not be consistent shifts in one direction, nor reproducible. For example, high variance of a measured analyte may be indicative of a condition, such that any extreme value is predictive of the condition. Such an analyte may have a high importance by SHAP, but poor statistical significance. Additionally, although p-values are normally arbitrarily thresholded, they are continuous values and can be treated as such. In the ML literature, feature attribution is also moving toward statistical measures, such as with the introduction of “knockoffs” to calculate false-discovery rates among features (https://candes.su.domains/publications/downloads/FDR_regression.pdf).

We agree with the reviewer here and have revised this portion of the analysis accordingly. Using SHAP to select important proteins is indeed complimentary to differential expression analysis. Our goal here was to showcase that GPS and large spectral libraries could be used to find potential panels of biomarkers that are successful in differentiating between subphenotypes of AKI. The language in the text has been clarified to reflect this in the Application subsection of the Results.

"Using the increased analytical depth of GPS, we applied recursive feature elimination (RFE) with cross validation via explainable artificial intelligence (SHAP [34]) and an XGBoost classifier to identify a panel of highly accurate predictive proteins in differentiating between AKI subphenotypes. SHAP-RFE analysis identified a group of 18 proteins as the most effective discriminatory panel(Figure5D),

Additionally, we instead perform recursive feature elimination with SHAP as the indicator of feature importance to select this panel of biomarkers to showcase. We found our explanation and justification of using raw SHAP scores to pick the most important proteins arbitrary, so we wanted to implement this in a cleaner manner in order to help streamline the story of the manuscript. The updated analysis is visible in Figure 5D-F.

Minor Issues

1. The opening sentence of the introduction states, "One disadvantage of data independent acquisition (DIA) proteomics is that a spectral library based on previously identified peptides in a sample is required to interpret the complex signal and quantify peptides." This is not strictly true, as there exist several library free approaches; for example, those that generate pseudo-spectra from correlated features, such as with DIAUmpire. Indeed, MSFragger-DIA is mentioned later in the manuscript.

We agree that this is not strictly true, so this sentence has been removed from the introduction.

2. Should "peakgroup" be a single word or two ("peak group")? My inclination is the latter.

Peak group has been replaced by Precursor throughout the text.

3. When describing the class imbalance problem with Percolator, the manuscript states, "... some true targets are identified, they would represent a small fraction compared to the negative decoys in the data, creating a overwhelming class imbalance, which can destabilize the training of machine learning algorithms if not dealt with in an appropriate manner." This statement is indeed true. Percolator, specifically, attempts to address this by selecting the cost hyperparameter for each class separately over a grid search. This essentially functions as a class weight parameter, which can downweight importance of the negative, decoy class relative to the positive, target class. Note that the nested cross-validation procedure is described by Granholm et al.

<https://bmcbioinformatics.biomedcentral.com/articles/10.1186/1471-2105-13-S16-S3>

This is a good point, and we have some text to the discussion to clarify how class imbalances can be handled and how we avoid these issues using GPS. The text is as follows:

"..but when the true target to decoy ratio is small the selected targets may be especially noisy, leading to a class imbalance. There are different ways to try and correct training set imbalances, such as down-sampling the majority class or up-sampling the minority class [25], and it is possible to provide the class ratios to certain machine learning algorithms to ensure that over-represented classes do not dominate the training loops. In the case of GPS, instead of choosing to implement imbalanced learning methods with sample specific classifiers we trained models on curated data that generalize to unrelated samples, so that it does not matter if there is a class imbalance or noisy labels in the data being predicted and score. To ensure that the no class imbalance issues effect training, we calculate the class ratios and pass them to our training algorithms so that weights are adjusted accordingly and the imbalance is taken into account during training."

REVIEWERS' COMMENTS:

Reviewer #1 (Remarks to the Author):

The authors did a great job in revising the manuscript and, in some parts, substantially. Which data was used for training and testing is now much more evident. The benchmarks directly provide direct insight into the additional benefit of the method. I can recommend the publication in Communications Biology.

Reviewer #2 (Remarks to the Author):

I appreciate all the efforts the authors showed in this revision. I'm satisfied with their responses. Here, I only have one suggestion. Do they provide the source codes for training GPS model? For bioinformatics researchers or software developers, they may would like to re-train the model using their own data. I think this should be provided.

Reviewer #3 (Remarks to the Author):

The authors have answered my questions and I am satisfied with the revised manuscript. Interesting work!

All source code for GPS is free and open source, and available on GitHub (<https://github.com/InfectionMedicineProteomics/gps>). We have included a command line option to train your own model, and the predict/scoring command line options can take in separately trained models and their scalers as parameters so any type of model trained using the same features can be used. There is documentation in the README file of the GPS GitHub repository that shows how to train your own model using the command line. Additionally, you can use the Python API to train your own models and apply it to new data.